# MATCHA: MULTI-STAGE RIEMANNIAN FLOW MATCHING FOR ACCURATE AND PHYSICALLY VALID MOLECULAR DOCKING

## ABSTRACT

Accurate prediction of protein-ligand binding poses is crucial for structure-based drug design, yet existing methods struggle to balance speed, accuracy, and physical plausibility. We introduce MATCHA, a novel molecular docking pipeline that combines multi-stage flow matching with learned scoring and physical validity filtering. Our approach consists of three sequential stages applied consecutively to progressively refine docking predictions, each implemented as a flow matching model operating on appropriate geometric spaces ($\mathbb{R}^3$, $SO(3)$, and $SO(2)$). We enhance the prediction quality through a dedicated scoring model and apply unsupervised physical validity filters to eliminate unrealistic poses. Compared to various approaches, MATCHA demonstrates superior performance on ASTEX and PDBBIND test sets in terms of docking success rate and physical plausibility. Moreover, our method works approximately $25\times$ faster than modern large-scale co-folding models.

## 1 INTRODUCTION

Molecular docking aims to predict the binding pose of a small molecule (ligand) within the active site of a target protein. It plays a key role in computer-aided drug discovery, particularly in virtual screening, the computational search for promising drug candidates within large-scale compound libraries. Given the vast size of these libraries, practical docking methods must balance accuracy with computational efficiency. Additionally, predicted poses are expected to be physically plausible (Buttenschoen et al., 2024). Another challenge is the diversity of existing docking benchmarks, which differ substantially in target and ligand selection, making it difficult to design methods that generalize well across all datasets.

Classical docking approaches (Friesner et al., 2004; Trott & Olson, 2010; Koes et al., 2013; Forli et al., 2016; Sulimov et al., 2020) have traditionally relied on hand-crafted scoring functions combined with heuristic search algorithms. However, recent benchmarks (Morehead et al., 2025) demonstrate that such methods are outperformed by data-driven approaches.

Modern data-driven blind docking methods (Abramson et al., 2024; Boitreaud et al., 2024; Wohlwend et al., 2024), starting from the seminal DIFFDOCK (Corso et al., 2022), typically formulate molecular docking as a generative modeling problem, where a neural network — often a diffusion model — learns to sample ligand poses from a probabilistic distribution.

Our proposed method, MATCHA, follows this generative paradigm but is based on flow MATCHing (Lipman et al., 2022) rather than diffusion. Following DIFFDOCK, we represent ligand flexibility in a joint space of translations, global rotations, and internal torsions. This corresponds to a semi-flexible ligand: the conformation is fixed except for rotations around rotatable bonds. In contrast to Riemannian diffusion-based methods, Riemannian flow matching (Chen & Lipman, 2023) provides tractable losses in these spaces and simplifies training. Moreover, our approach naturally bypasses the need for semi-flexible conformational alignment, which is a challenging optimization problem. To the best of our knowledge, it is the first docking pipeline that is built upon flow matching on non-Euclidean manifolds.

Architecturally, MATCHA combines the structure of a Diffusion Transformer (DIT; Peebles & Xie 2023) with a spatial encoder inspired by UNIMOL (Zhou et al., 2023). Our pipeline consists of three neural networks trained at different noise levels. For instance, the first model is optimized to predict the 3D translational displacement of the ligand center relative to the binding site. By default, MATCHA is trained for blind docking, but the pocket-informed setting can be managed by omitting the coarse model and providing the correct binding site location. Also, we train a separate scoring model to select the best pose among all generated candidates.

Our main contributions are as follows:

- We introduce MATCHA, a neural pipeline for molecular docking that combines Riemannian flow matching with a DIT-inspired architecture. The pipeline employs three neural networks applied consecutively to refine docking predictions progressively.

- We perform an extensive empirical evaluation of MATCHA against state-of-the-art methods, comparing binding quality, computational efficiency, and physical plausibility across the PDBBIND, DOCKGEN, ASTEX, and POSEBUSTERS V2 benchmarks.

- MATCHA achieves inference approximately $25\times$ faster than ALPHAFOLD 3, CHAI-1, and BOLTZ-2, while having state-of-the-art docking performance on the ASTEX test set: 66% with $\text{RMSD} \leq 2\,\text{Å}$ & PB-valid (Buttenschoen et al., 2024) and competitive results on other benchmarks.

## 2 METHOD

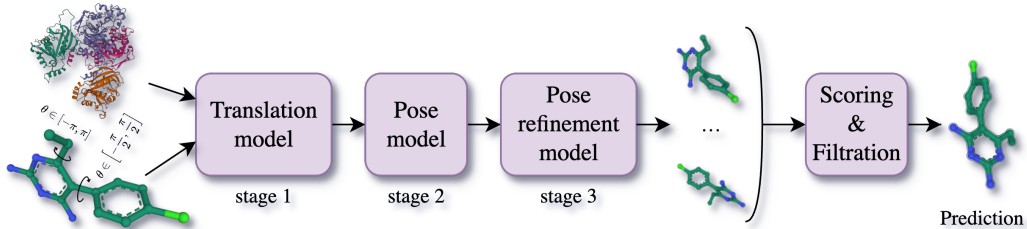

**Figure 1:** MATCHA consists of three flow matching models that generate a set of ligand poses. After unsupervised physical validity filtration, all poses are scored with a separate model, and the prediction is the pose with the best score. We highlight rotatable bonds and their periods in the ligand on this figure.

### 2.1 DOCKING LOSS FUNCTION

MATCHA tackles the molecular docking problem by modeling a protein as a rigid body while parameterizing the ligand's degrees of freedom in a manner similar to DIFFDOCK. Specifically, we operate in the following spaces:

- **translation (tr)**: a 3D continuous vector representing the position of the ligand's center relative to the protein,

- **rotation (rot)**: an $\text{SO}(3)$ transformation matrix representing the orientation of ligand,

- **torsion angles (tor)**: a set of angles in $\text{SO}(2)$, one for each rotatable bond in the ligand.

For rotatable bonds, we define the torsional period $p$ as the smallest positive angle such that a rotation by $p$ about that bond yields an indistinguishable configuration (up to symmetry). Accordingly, torsion angles are taken modulo $p$, i.e., for any $\theta$ we use its wrapped representative $\theta \bmod p$ that is in $(-p/2,\, p/2]$.

Our model predicts velocities $\boldsymbol{v}_{\text{tr}}$, $\boldsymbol{v}_{\text{rot}}$, and $\boldsymbol{v}_{\text{tor}}$ in the tangent spaces $\mathbb{R}^3$, $\mathfrak{so}(3)$, and $\mathfrak{so}(2)^m$, with $m$ denoting the number of rotatable bonds. Elements of $\mathfrak{so}(n)$ can be thought of as $n \times n$ real skew-symmetric matrices (Warner, 1983), we represent them as $n(n-1)/2$-dimensional vectors.

We compute separate flow matching losses (Chen & Lipman, 2023) for each component and optimize their weighted sum. Generally, to train a flow matching model $v_\theta$ given data points $x_1$ on a Riemannian manifold $\mathcal{M}$ drawn from $p_{\text{data}}$, we should define a noise distribution $p_0$. Then we choose an interpolation on $\mathcal{M}$, select an appropriate norm, and compute the time derivative of the interpolation to obtain the conditional velocity. The conditional flow matching loss takes the form:

$$\mathcal{L}_{\text{CFM}} = \mathbb{E}_{x_0 \sim p_0, x_1 \sim p_{\text{data}}, t \sim U[0,1]} \|v_\theta(x_t, t) - \dot{x}_t\|, \text{ where } x_t = \text{interpolate}(x_0, x_1; t). \quad (1)$$

For translations, we use a standard linear interpolation and normally distributed noise. For both angular components, we adopt spherical linear interpolation (SLERP; Shoemake 1985). In SO(3), the time derivative of SLERP is computed by transforming to quaternion representations and applying automatic differentiation with custom backward functions. Additional derivation details are provided in Appendix A.

## 2.2 ARCHITECTURE

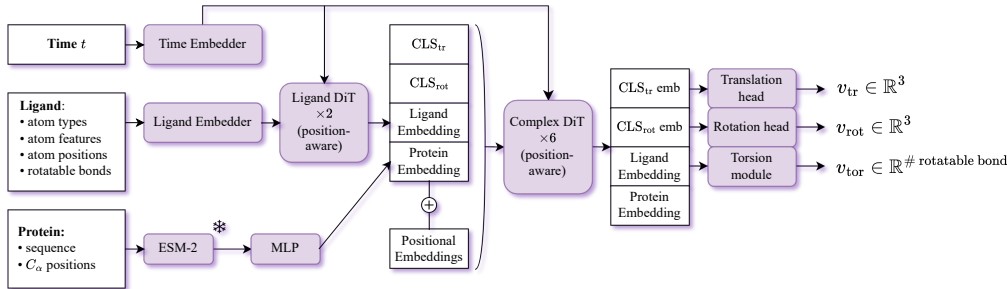

**Figure 2:** The architecture of the velocity model of MATCHA (stages 1, 2, 3).

MATCHA consists of two primary components (Figure 1): the docking pipeline and the scoring model. These components are implemented through a transformer-based architecture and have a similar design (Figure 2).

### 2.2.1 VELOCITY MODEL

**Input tokens** The input sequence consists of ligand atom tokens, protein amino acid residue tokens, and two CLS-like tokens for aggregating translation ($\text{CLS}_{\text{tr}}$) and global rotation ($\text{CLS}_{\text{rot}}$) information. Each token is assigned a 3D coordinate: atom positions for ligand atoms, $C_\alpha$ positions for residues, and the ligand centroid for both CLS tokens. Protein representations are initialized from ESM-2-35M (Lin et al., 2022) embeddings; we use the 35M model instead of the 650M variant to reduce the risk of overfitting. Initial ligand atom embeddings are a sum of simple embeddings of scalar and categorical atom features from the RDKit package (Landrum, 2024).

Time $t \in [0, 1]$ is embedded by an MLP over sinusoidal features and conditions all transformer blocks in a DiT-like manner. Positions $(x, y, z)$ are encoded using a simple MLP and added to both embeddings of ligand atoms and protein residues in a manner of positional encoding in transformers.

**Distance-aware attention bias** We adopt the approach from UNIMOL and ALPHAFOLD 3, where spatial features are used as extra biases in self-attention. Given 3D coordinates $x = \{x_i\}_{i=1}^N$, $x_i \in \mathbb{R}^3$, we form a per-head attention bias by combining a radial (distance-based) and a directional (vector-based) term. For a pair $(i, j)$ with edge type $t_{ij} \in \{1, \dots, T\}$, define the displacement $\Delta_{ij} = x_i - x_j$ and a stabilized inverse distance

$$s_{ij} = \frac{1}{\|\Delta_{ij}\|_2^2 + 1}. \quad (2)$$

An edge-type–specific affine transform produces $\tilde{s}_{ij} = \alpha_{t_{ij}} s_{ij} + \beta_{t_{ij}}$. We then embed $\tilde{s}_{ij}$ with a $K$-kernel Gaussian RBF:

$$\phi_{ij}^{(k)} = \mathcal{N}(\tilde{s}_{ij}; \mu_k, \sigma_k^2), \qquad k = 1, \dots, K, \quad (3)$$

followed by a small MLP projection $g : \mathbb{R}^K \to \mathbb{R}^H$ to obtain a per-head radial bias $o_{ij} = g(\phi_{ij}) \in \mathbb{R}^H$. In parallel, a directional projection $h : \mathbb{R}^3 \to \mathbb{R}^H$ maps the displacement to $v_{ij} = h(\Delta_{ij})$. The per-pair bias is $b_{ij} = g(\phi_{ij}) + h(\Delta_{ij}) \in \mathbb{R}^H$. Stacking over all pairs yields a tensor in $\mathbb{R}^{N \times N \times H}$; we then move the head dimension to the front to obtain $B_{hij} = [b_{ij}]_h \in \mathbb{R}^{H \times N \times N}$, which we add to the attention logits.

**Velocity prediction heads**    After the transformer backbone, we employ lightweight modules to predict the velocity fields. We do not force rotational or translational symmetries via the architecture; instead, we rely on data augmentations during training to promote invariances and equivariances.

The translation head consumes the dedicated $\mathrm{CLS}_{\mathrm{tr}}$ token and outputs a 3D velocity vector $v_{\mathrm{tr}} \in \mathbb{R}^3$. The rotation head consumes the $\mathrm{CLS}_{\mathrm{rot}}$ token and outputs a 3-vector representation of $\mathfrak{so}(3)$. For torsions, we construct a token for each rotatable bond by averaging the embeddings of ligand atoms influenced by the rotation of that bond and combining this with an encoding of basic features of the bond level. The resulting per-bond sequence is passed through a lightweight transformer decoder, and a final single-layer MLP projects each token to a scalar torsional velocity $v_{\mathrm{tor}}$.

**Coarse-to-fine structure**    Our pipeline stacks three models of identical architecture but independent weights. The first model is used solely for translation, where samples are drawn from a zero-mean Gaussian distribution with a large variance, while angular components are sampled uniformly. The second model refines translation using a Gaussian centered at the ground truth with moderate variance, still keeping angular components uniform. Finally, the third model sharpens both translation and angular degrees of freedom, sampling them from Gaussians with small variance around the ground truth. The models are trained independently. Full details are provided in Algorithm 1.

The system can operate in two distinct scenarios: blind docking and pocket-aware docking. Blind docking means predicting ligand poses without prior knowledge of the binding site location, while in the pocket-aware scenario, the information about the known binding site is used to guide pose prediction. This flexibility is achieved because of the multiscale nature of MATCHA.

**Augmentations**    During training, we apply multiple augmentation techniques to avoid overfitting and improve model generalization. Firstly, we randomly rotate the whole complex to get new $(x, y, z)$ coordinates. Secondly, we add random Gaussian noise with zero mean and standard deviation 0.25 to the protein and ligand positions. Finally, we randomly mask 15% of protein residues and ligand atoms. This strategy also leads to masking of some rotatable bonds.

**Inference**    We run a fixed-length explicit Riemannian Euler solver (10 steps) over $(\mathrm{tr}, \mathrm{rot}, \mathrm{tor})$ using the predicted velocities, applying three models sequentially.

Stage 1    From a random initialization, we integrate all degrees of freedom (translation, rotation, torsions), but retain only the predicted translation; the angular components are discarded and reinitialized uniformly.

Stage 2    Starting from this state (predicted translation and uniformly distributed angles), we perform the same rollout and pass the full output $(\mathrm{tr}, \mathrm{rot}, \mathrm{tor})$ forward.

Stage 3    We execute the final rollout to produce the refined pose.

### 2.2.2    POSE SELECTION

**Scoring model**    We train a separate pose–scoring network to evaluate and rank candidate docking poses. It shares the backbone with our docking model but removes time conditioning and all flow matching components. Instead, a dedicated scalar scoring head is optimized with an RMSD-based pairwise ranking objective for comparative pose assessment. For training, each batch is composed of multiple noisy poses of the same protein–ligand complex, and the model learns to order the resulting pairs of poses.

**Pose filtration**    We reimplement, speed up, and apply a minimal set of PoseBusters geometric and physicochemical validity filters before scoring. Specifically, we retain candidate complexes that

---

**Algorithm 1:** General scheme of MATCHA docking training

---

**Input:** Protein-ligand complexes $D$, initialized flow model $v$, stage $\in \{1, 2, 3\}, \sigma_{\text{large}}, \sigma_{\text{medium}}, \sigma_{\text{small}}$

**for** *batch of ligand-protein complexes* **in** $D$ **do**

     1. Take conformation of ligand from batch. Compute the centroid $\text{tr}_{\text{true}}$ of this conformation. Apply augmentations (Section 2.2.1, augmentations).

     2. Identify rotatable bonds and define their quantity by $m$.

     3. Sample random translation, rotation, and torsion: $\text{tr}, \text{rot}, \text{tor}$ and apply them to the conformation.

     4. Set $\text{rot}_{\text{true}} := \text{inverse}(\text{rot}), \text{tor}_{\text{true}} := \text{inverse}(\text{tor})$.

     5. Sample noise for translation, rotation, and torsion transformations $\text{tr}_{\text{noise}}, \text{rot}_{\text{noise}}, \text{tor}_{\text{noise}}$:

       • if stage $= 1$: $\text{tr}_{\text{noise}} \sim \mathcal{N}(0, \sigma_{\text{large}}^2), \text{rot}_{\text{noise}} \sim \text{Unif}(\text{SO}(3)), \text{tor}_{\text{noise}} \sim \text{Unif}(\text{SO}(2))^m$

       • if stage $= 2$: $\text{tr}_{\text{noise}} \sim \mathcal{N}(\text{tr}_{\text{true}}, \sigma_{\text{medium}}^2), \text{rot}_{\text{noise}} \sim \text{Unif}(\text{SO}(3)), \text{tor}_{\text{noise}} \sim \text{Unif}(\text{SO}(2))^m$

       • if stage $= 3$: $\text{tr}_{\text{noise}} \sim \mathcal{N}(\text{tr}_{\text{true}}, \sigma_{\text{small}}^2), \text{rot}_{\text{noise}} \sim \mathcal{N}(\text{rot}_{\text{true}}, \sigma_{\text{small}}^2 \boldsymbol{I}), \text{tor}_{\text{noise}} \sim \mathcal{N}(\text{tor}_{\text{true}}, \sigma_{\text{small}}^2)^m$

     6. Sample $t \sim \text{Uniform}(0, 1)$, interpolate transformations between noisy $(\text{tr}_{\text{noise}}, \text{rot}_{\text{noise}}, \text{tor}_{\text{noise}})$ and true values $(\text{tr}_{\text{true}}, \text{rot}_{\text{true}}, \text{tor}_{\text{true}})$, and apply these transformations to the ligand conformation, resulting in $\boldsymbol{x}(t) := (\text{tr}(t), \text{rot}(t), \text{tor}(t))$.

     7. Obtain the output of the flow model, $v(\boldsymbol{x}(t), t)$, which belongs to $\mathbb{R}^3 \times \mathfrak{so}(3) \times \mathfrak{so}(2)^m$.

     8. Compute the flow matching loss for each component of $v(\boldsymbol{x}(t), t)$ separately and compute their linear combination.

     9. Execute the gradient optimization step of the computed loss.

**Output:** Trained flow model $v$ for the given stage

---

achieve the highest filter scores across key validity criteria: (i) *Minimum distance to protein*, preventing ligand–receptor atomic collision; (ii) *Protein–ligand maximum distance*, excluding poses with excessive protein–ligand separation that are unlikely to form specific interactions; (iii) *Volume overlap with protein*, rejecting any nonzero volumetric overlap with the receptor; and (iv) *Internal steric clash*, removing ligand conformers with intramolecular clashes. Filtration is strictly unsupervised—i.e., it does not rely on knowledge of the native pose – and is therefore readily applicable at inference. In practice, for each complex we retain the poses that pass the highest number of validity criteria and then select a single pose using our learned scoring model.

## 3 EXPERIMENTAL SETUP

### 3.1 DATASETS

MATCHA is trained on two major protein-ligand complex datasets: PDBBIND (Liu et al., 2017) and BINDING MOAD (Hu et al., 2005). During training, we keep only protein chains close to the ligand (less than $4.5$ Å). For both datasets, we use splits provided by (Corso et al., 2024). Additional MOAD complexes belong to pocket clusters of PDBBIND train set. We use complexes that have proteins with less than 2000 residues and ligands with $6 - 150$ heavy atoms. At training time, we concatenate the BINDING MOAD dataset to the PDBBIND training set without removing redundant complexes, thereby giving additional weight to higher-quality complexes that passed both datasets' filtering processes.

For the BINDING MOAD dataset, we implement protein-level sampling to address the inherent class imbalance where multiple ligands are bound to the same protein structure. During every training epoch, we sample each unique protein exactly once. For each selected protein, we randomly choose one ligand from all ligands bound to that receptor.

The docking quality is evaluated on four test datasets. ASTEX Diverse set (Hartshorn et al., 2007) and POSEBUSTERS V2 Benchmark set (Buttenschoen et al., 2024) are commonly used in the field and contain 85 and 308 complexes, respectively. DOCKGEN (Corso et al., 2024) is a set of 330 hard complexes with binding sites different from the training set. Finally, we use the PDBBIND test set obtained using time-splitting of dataset complexes and has 363 complexes (Corso et al., 2022).

## 3.2 Training details

All models are trained with the AdamW optimizer (Loshchilov & Hutter, 2017) using a learning rate of $5 \times 10^{-5}$. The docking models employ a batch size of 24 and are trained on a single NVIDIA H100 GPU (80GB). Stage 1 training runs for 1.9M steps ($\approx$ 11 days), stage 2 for 3.3M steps ($\approx$ 19 days), and stage 3 for 1.4M steps ($\approx$ 8 days), totaling 38 GPU-days. The scoring model is trained separately with batch size 12 on an NVIDIA V100 GPU (16GB) for 700k steps ($\approx$ 26 hours).

All MATCHA models use 6 Complex DIT layers with hidden dimension 192. This corresponds to $\sim$29M parameters for the docking models and $\sim$6M parameters for the scoring model. For docking, we optimize a weighted objective with loss coefficients $w_{\text{tr}} = 1$, $w_{\text{rot}} = 1$, and $w_{\text{tor}} = 3$.

## 3.3 Docking Quality Metrics

To comprehensively evaluate the performance of MATCHA, we use symmetry-corrected Root Mean Square Deviation (RMSD) that accounts for molecular symmetry and report success rates at 2 Å threshold (RMSD $\leq$ 2 Å). We also run POSEBUSTERS tests (Buttenschoen et al., 2024) to assess physical plausibility of predicted poses, resulting in the combined metric RMSD $\leq$ 2 Å & PB-valid. Exact POSEBUSTERS tests are listed in Appendix I. For models that predict whole complex structures, we follow Abramson et al. (2024) and use pocket-aligned symmetric RMSD by aligning the reference protein pocket to the predicted structure. If a model crashes on a complex, we assign an RMSD of $+\infty$. The details are reported in Section 4.5 and Appendix G.1.

## 3.4 How we run baselines

**Baseline model parameters**  For ALPHAFOLD 3 (Abramson et al., 2024), BOLTZ-2 (Passaro et al., 2025) and CHAI-1 (Boitreaud et al., 2024), we use the prediction with the highest confidence score among five model seeds with five samples per seed and 10 recycling steps. We used multiple sequence alignments computed with JACKHMMER (HMMER3) (Eddy, 2011). When modeling receptor structures, we preserve all chains and remove exact duplicate chains only. NEURALPLEXER (Qiao et al., 2024b), DIFFDOCK-L (Corso et al., 2024) and UNI-MOL (Alcaide et al., 2024) are run with their default inference parameters. We run FLOWDOCK (Morehead & Cheng, 2025) using true holo protein structures as receptor templates. For all methods, we take the top-scored sample among generated. We detail the parameters that were used to run classical docking models (AUTODOCK VINA (v1.2.5) (Trott & Olson, 2010), SMINA (v2020.12.10; fork of Vina 1.1.2) (Koes et al., 2013), and GNINA (v1.0.3) (McNutt et al., 2021)) in Appendix B.

**Pocket Detection for Classical Docking Methods**  Classical docking methods and UNI-MOL face fundamental challenges in blind docking due to their reliance on pocket identification algorithms. This dependence introduces additional error when binding sites are unknown, representing a key limitation compared to end-to-end methods. We evaluate these methods using P2RANK (Krivák & Hoksza, 2018), FPOCKET (Le Guilloux et al., 2009), and whole protein approaches. Among these, P2RANK has shown the best performance and is used as the primary pocket detection method for our main results. The complete results are provided in Appendix C.1.2.

**MATCHA inference parameters**  We run MATCHA in two inference regimes: in blind docking setup using all protein chains and in the pocket-aware scenario with the correct binding site provided. The first model in the pipeline generates 40 samples, which are then passed to the refinement stages. Then, the plausibility filtering is applied, reducing the number of candidate poses. We use the top-scored pose among the rest of the candidates.

## 4 Results and Discussion

### 4.1 Comprehensive Evaluation on Diverse Benchmarks

We evaluate MATCHA against established baselines: classical docking (SMINA, VINA, GNINA), deep learning-based (DL-based) methods (DIFFDOCK, UNI-MOL, NEURALPLEXER and FLOW-DOCK) as well as co-folding models (ALPHAFOLD 3, CHAI-1, BOLTZ-2). The results for DockGen

dataset as well as all extended results are reported in Appendix C.1. Additionally, we evaluate the considered models in the pocket-aware scenario with the corrected binding site location provided (see Appendix C.2).

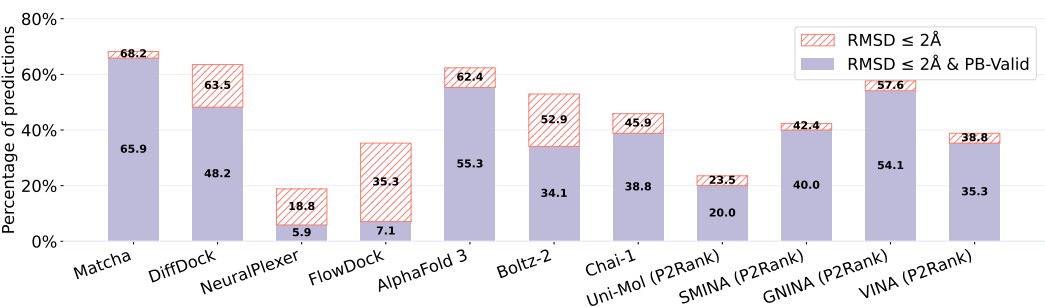

**Figure 3:** Blind ligand docking success rates on ASTEX Diverse Set ($n = 85$).

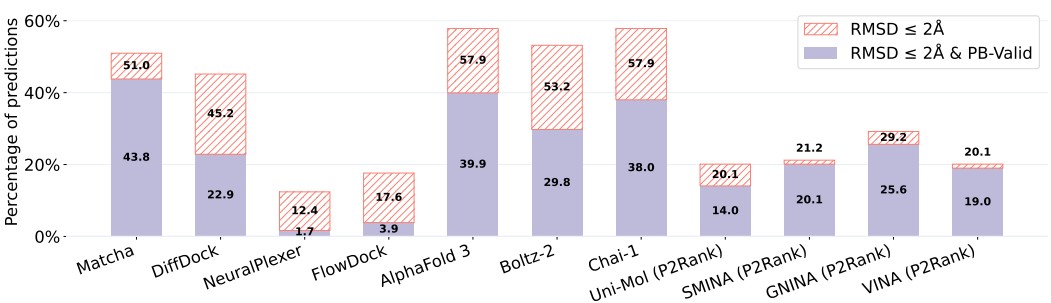

**Figure 4:** Blind ligand docking success rates on PDBBIND test set ($n = 363$).

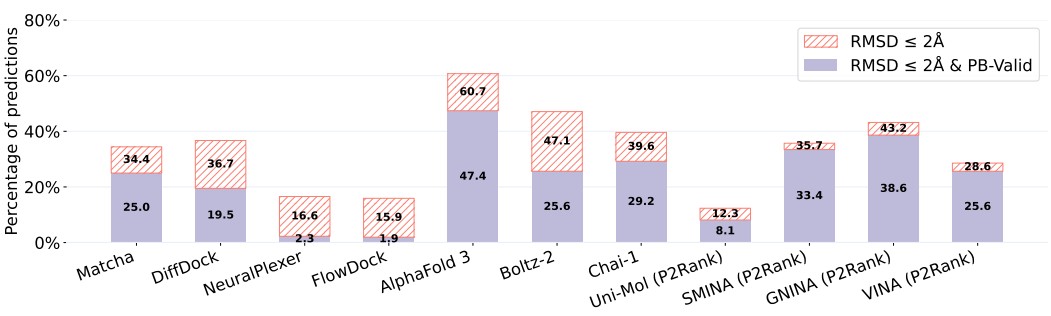

**Figure 5:** Blind ligand docking success rates on POSEBUSTERS V2 dataset ($n = 308$).

On the ASTEX dataset, MATCHA demonstrates superior performance across all metrics, achieving the highest success rates for both RMSD $\leq 2$ Å (68.2%) and the harder RMSD $\leq 2$ Å & PB-valid (65.9%), outperforming the next-best method (ALPHAFOLD 3) by 10.6 percentage points on physically valid structures. The substantial improvement in the number of physically valid structures highlights the effectiveness of our flow matching approach in generating chemically plausible poses while maintaining high geometric accuracy.

On the PDBBIND test set, MATCHA demonstrates superior performance across all metrics compared to DL-based docking methods. While co-folding methods achieve slightly higher RMSD $\leq 2$ Å success rates, MATCHA shows the highest success rate of 43.8% for RMSD $\leq 2$ Å & PB-valid, surpassing even ALPHAFOLD 3 (39.9%). This demonstrates that MATCHA produces the most physically plausible and chemically valid structures, making it particularly valuable for practical drug discovery applications.

The POSEBUSTERS V2 dataset presents the most challenging evaluation scenario, where MATCHA's performance shows a relative decrease compared to co-folding methods due to a higher proportion of protein targets with binding pockets structurally dissimilar to the training data (Morehead et al., 2025). Co-folding methods, trained on significantly larger and more diverse protein structure datasets, demonstrate better generalization to these out-of-distribution pocket geometries. Nevertheless, MATCHA maintains superior performance compared to other DL-based methods (DIFFDOCK-L, NEURALPLEXER, FLOWDOCK), confirming the effectiveness of our approach within the shared training distribution. This evaluation highlights the fundamental limitation of such models in generalizing beyond training data distributions.

## 4.2 COMPUTATIONAL EFFICIENCY OF MATCHA

**Inference Speed Analysis** We measure the average inference time for all considered blind docking methods one NVIDIA A100 40Gb GPU (see Figure 6). Time is measured only for model inference avoiding model loading. The exact timing results (in seconds) are detailed in Appendix H. MATCHA demonstrates the best speed-accuracy balance in terms of fraction of PoseBusters-valid predictions with $\mathrm{RMSD} \leq 2\,\text{Å}$, which is important in practical applications. MATCHA achieves substantially higher docking success rates compared to fast methods (DIFFDOCK-L, NEURALPLEXER, FLOW-DOCK), having even faster inference time. Also, MATCHA shows a significant speed advantage over high-accuracy co-folding methods, achieving comparable docking quality.

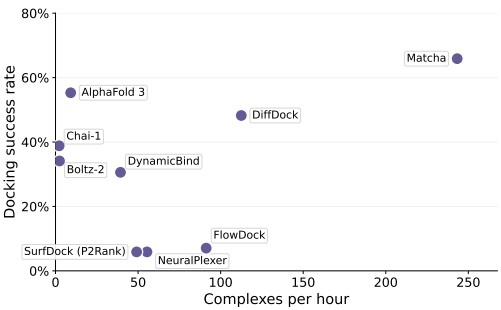

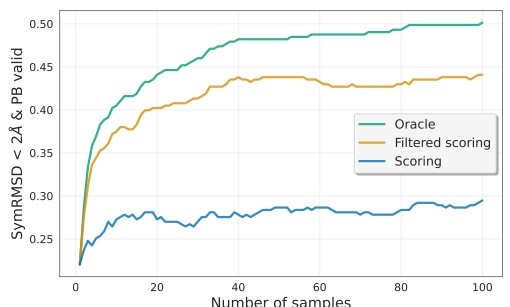

**Figure 6:** The dependence between the average docking inference time and percentage of PoseBusters-valid predictions for ASTEX dataset.

**Figure 7:** The dependence between the number of generated samples and MATCHA docking quality ($\mathrm{RMSD} \leq 2\,\text{Å}$ & PB valid) for PDBBIND test set.

**Training speed** MATCHA required significantly less computational resources for training compared to competing methods: 38 GPU-days on a single H100 versus 120 GPU-days for DIFFDOCK-L (4× RTX A6000) and 3,840 GPU-days for CHAI-1 (128× A100). This highlights MATCHA's efficiency in both training and inference compared to DL-based models and co-folding models.

## 4.3 ANALYSIS OF THE REQUIRED NUMBER OF SAMPLES

Figure 7 shows how the number of generated poses affects the quality of the best selected pose, comparing scoring-only regime, scoring with filtration (default), and oracle performance (representing the theoretical upper bound). Using scoring model increases $\mathrm{RMSD} \leq 2\,\text{Å}$ & PB valid from 0.22 to 0.27, while adding physically-aware post-filtration improves results to 0.44, demonstrating the critical importance of incorporating molecular validity constraints alongside learned scoring functions. Performance plateaus at around 40 samples, marking the optimal computational cost-quality trade-off. The gap between filtered scoring and oracle performance indicates remaining opportunities for improvement in pose selection, while the convergence behavior suggests that the current sampling strategy effectively explores the relevant conformational space within the first 40 generated poses.

## 4.4 PHYSICAL VALIDITY ASSESSMENT

A critical advantage of MATCHA lies in its ability to generate physically plausible molecular poses. For instance, on the PDBBIND test set, MATCHA has 85.9% of physically-valid structures among

those with RMSD $\leq 2\,\text{Å}$. To compare, ALPHAFOLD 3 has 68.9%, DIFFDOCK-L – 50.7%. This consistent superiority in generating chemically valid poses across different datasets demonstrates that MATCHA's flow matching approach effectively preserves molecular constraints through two key mechanisms: (1) ligand parametrization using only torsional angles, which maintains internal molecular geometry, and (2) fast and effective post-filtration to eliminate unrealistic complex poses. The high fraction of physically plausible structures, combined with fast inference, makes MATCHA particularly suitable for practical drug discovery applications where chemical validity and screening efficiency are as important as geometric accuracy.

### 4.5 IMPACT OF ALIGNMENT STRATEGY ON REPORTED METRICS

Pocket alignment refers to the procedure for superimposing predicted and reference protein–ligand complexes prior to RMSD evaluation, which is necessary for models that modify protein structure during prediction. Different strategies lead to systematically different results. One common option is the POCKET-BASED alignment, where the predicted pocket is aligned to the reference one. This can yield lower RMSD values even if the ligand is placed in a non-native binding site, effectively inflating success rates for both structure prediction and rigid docking methods. In contrast, we follow the BASE alignment strategy, in which the reference pocket is aligned to the full predicted protein, ensuring that incorrect pocket assignments are penalized. This methodological difference explains why our reported metrics may differ from those in other studies, even when using the same datasets. Appendix G provides a detailed comparison, showing that pocket-based alignment consistently increases success rates by 10–20% across methods, while leaving their relative ordering unchanged.

## 5 RELATED WORK

Prior work on molecular docking can be organized along several orthogonal axes. One natural division is between classical heuristic-based approaches and modern deep learning methods. Within these, methods can be further distinguished by whether they treat docking as rigid-body alignment or as a co-folding process, and by whether they rely on discriminative or generative modeling of the complex.

Co-folding approaches represent some of the most computationally demanding directions in molecular docking, as they attempt to jointly predict the conformations of both proteins and ligands. Recent examples include ALPHAFOLD 3 (Abramson et al., 2024), CHAI-1 (Boitreaud et al., 2024), and BOLTZ-1/2 (Wohlwend et al., 2024; Passaro et al., 2025), NEURALPLEXER family (Qiao et al. 2024b, Qiao et al. 2024a), INTERFORMER (Lai et al., 2024), DYNAMICBIND (Lu et al., 2024), LABIND (Zhang et al., 2025b), PHYSDOCK (Zhang et al., 2025a). These models are generally diffusion generative models in the Euclidean space. Since co-folding methods model protein positions, they typically require large-scale training and long inference time.

In contrast, rigid docking methods assume a fixed protein conformation and focus on placing a flexible ligand into binding site. This setting is computationally simpler than co-folding, yet it remains challenging due to the high dimensionality of ligand torsions and the rugged energy landscape of protein pockets. Classical docking approaches, such as AUTODOCK VINA (Trott & Olson, 2010) and SMINA (Koes et al., 2013), rely on heuristic search combined with hand-crafted scoring functions. Recent deep learning methods reformulate rigid docking either as a regression problem or as generative modeling. Regression-based models, including EQUIBIND (Stärk et al., 2022), TANKBIND (Lu et al., 2022), E3BIND (Zhang et al., 2022), and FABIND (Pei et al., 2023), UNI-MOL (Alcaide et al., 2024), predict a single pose in one shot, often followed by torsional refinement of the ligand. Generative methods, such as DIFFDOCK (Corso et al., 2022) and FLOW-DOCK (Morehead & Cheng, 2025), instead learn distributions over poses and can sample diverse ligand conformations conditioned on the rigid receptor.

## 6 CONCLUSION

We introduced MATCHA, a multi-stage Riemannian flow matching framework for molecular docking that combines geometric generative modeling, scoring, and physical validity filtering. MATCHA

achieves a high physically valid docking accuracy while being substantially faster than many other models, making it suitable for large-scale applications. A key limitation is reduced generalization to unseen protein pockets, pointing to future work on receptor flexibility and broader protein coverage. Overall, MATCHA strikes a practical balance between accuracy, efficiency, and physical realism for structure-based drug discovery.

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

APPENDIX

## A    FLOW MATCHING ON $SO(2)$ AND $SO(3)$

In this section, we address the problem of training flow matching on the special orthogonal groups $SO(2)$ and $SO(3)$. Our approach involves minimizing the flow matching loss function with respect to the parameters $\boldsymbol{w}$ of a flow model denoted by $v$. The loss function is defined as:

$$\ell(\boldsymbol{w}) = \mathbb{E}_{\boldsymbol{x}_0, \boldsymbol{x}_1, t} \left\| v(\boldsymbol{x}(t), t; \boldsymbol{w}) - \frac{\mathrm{d}\boldsymbol{x}(t)}{\mathrm{d}t} \right\|_g, \tag{4}$$

In the equation above, $\boldsymbol{x}(t)$ represents an interpolation between two points $\boldsymbol{x}_0$ and $\boldsymbol{x}_1$. The term $\|\boldsymbol{x}\|_g = \sqrt{g(\boldsymbol{x}, \boldsymbol{x})}$ defines a norm based on the Riemannian metric $g$.

For a given Riemannian manifold $M$, the tangent space at a point $p \in M$ is denoted by $\mathcal{T}_p M$. The Riemannian metric is defined as:

$$g : \mathcal{T}_p M \times \mathcal{T}_p M \to \mathbb{R}, \tag{5}$$

acts on the Cartesian product of these tangent spaces to produce a non-negative scalar value.

The special orthogonal group $SO(n)$ consists of elements that can be represented as $n \times n$ rotation matrices. Although there are several ways to define a Riemannian metric $g$ on $SO(n)$, the canonical metric is given by:

$$g(\mathbf{X}, \mathbf{Y}) \triangleq \mathrm{tr}(\mathbf{X}^\top \mathbf{Y}), \tag{6}$$

where $\mathbf{X}$ and $\mathbf{Y}$ are $n \times n$ matrices corresponding to the elements of the tangent space.

Regarding interpolation on these manifolds, various methods exist. In this work, we employ the widely used spherical linear interpolation, commonly referred to as SLERP Shoemake (1985).

### A.1    $SO(2)$ MANIFOLD

Every $2 \times 2$ rotation matrix is characterized by the rotation angle $\theta$ and is given by

$$\mathbf{R} = \begin{bmatrix} \cos(\theta) & -\sin(\theta) \\ \sin(\theta) & \cos(\theta) \end{bmatrix}. \tag{7}$$

Given two rotation matrices defined by angles $\theta_0$ and $\theta_1$, the SLERP interpolation between them for $t \in [0, 1]$ is represented by the matrix $\mathbf{R}(t)$ with the angle

$$\theta(t) = \theta_0 + t(\theta_1 - \theta_0) = \theta_0 + t\Delta\theta. \tag{8}$$

The time derivative of this matrix is

$$\dot{\mathbf{R}}(t) = \begin{bmatrix} -\sin(\theta(t)) & -\cos(\theta(t)) \\ \cos(\theta(t)) & -\sin(\theta(t)) \end{bmatrix} \Delta\theta, \tag{9}$$

which can be expressed as

$$\dot{\mathbf{R}}(t) = \mathbf{R}(t) \begin{bmatrix} 0 & -\Delta\theta \\ \Delta\theta & 0 \end{bmatrix}. \tag{10}$$

This form — a product of a rotation matrix and a skew-symmetric matrix — can be derived for any element of $SO(n)$, given that the tangent space of an identity $SO(n)$ matrix is spanned by skew-symmetric $n \times n$ matrices.

Assuming the neural network model $\hat{v}$ for the flow takes as input $\theta$ and $t$, and outputs a scalar $\hat{v}(\theta, t)$, we can represent the flow as

$$v(\mathbf{R}(t), t) = \mathbf{R}(t) \begin{bmatrix} 0 & -\hat{v}(\theta(t), t) \\ \hat{v}(\theta(t), t) & 0 \end{bmatrix}. \tag{11}$$

Utilizing the canonical metric (6), the norm difference becomes

$$\|v(\mathbf{R}(t), t) - \dot{\mathbf{R}}(t)\| = \sqrt{2} \, |\theta_1 - \theta_0 - \hat{v}(\theta(t), t)| \, . \tag{12}$$

Consequently, our objective to be minimized is

$$\ell(\mathbf{w}) = \mathbb{E}_{\theta_0, \theta_1, t} \, (\theta_1 - \theta_0 - \hat{v}(\mathbf{R}(t), t, \mathbf{w}))^2 \, . \tag{13}$$

## A.2 $SO(3)$ MANIFOLD

In contrast to $SO(2)$ case, SLERP for $3 \times 3$ rotation matrices is more complicated, often requiring a transformation into quaternions (Brégier, 2021). Nonetheless, it is possible to compute the interpolated matrix $\mathbf{R}(t)$ and its corresponding time derivative, which belongs to the tangent space of $\mathbf{R}(t)$, using automatic differentiation tools.

Every tangent vector of a point $\mathbf{R} \in SO(3)$ can be expressed as

$$\mathbf{R} \begin{bmatrix} 0 & -k_z & k_y \\ k_z & 0 & -k_x \\ -k_y & k_x & 0 \end{bmatrix} . \tag{14}$$

Assuming that the values $k_x, k_y, k_z$ describe the matrix $\dot{\mathbf{R}}(t)$, a neural network can be constructed to yield three outputs: $v_x, v_y, v_z$. The flow model becomes

$$v(\mathbf{R}(t), t) = \mathbf{R}(t) \begin{bmatrix} 0 & -v_z & v_y \\ v_z & 0 & -v_x \\ -v_y & v_x & 0 \end{bmatrix} . \tag{15}$$

For the $SO(3)$ manifold, the square of norm in equation (4) evaluates to

$$2(k_x - v_x)^2 + 2(k_y - v_y)^2 + 2(k_z - v_z)^2 . \tag{16}$$

To determine the values $k_x, k_y, k_z$, one can leverage automatic differentiation to compute $\dot{\mathbf{R}}(t)$. Subsequently, these values can be extracted from the expression $\mathbf{R}^\top(t) \dot{\mathbf{R}}(t)$.

## B    HOW WE RUN CLASSICAL DOCKING

**Classical docking baseline parameters**    We used AUTODOCK VINA (v1.2.5) (Trott & Olson, 2010), SMINA (v2020.12.10; fork of Vina 1.1.2) (Koes et al., 2013), and GNINA (v1.0.3) (Mc-Nutt et al., 2021). Unless noted otherwise, all runs used `exhaustiveness 64` and `seed 42`. Pocket centers were provided as described below; the search box was centered at each pocket center and sized to the RDKit conformer diameter of the ligand plus a 10 Å padding on all six sides (equivalently, `autobox_add 10` where applicable). To emulate blind docking we also used a large box centered on the protein with padding `autobox_add 16`.

**Ligand and receptor preparation for classical docking**    Each ligand was prepared from a SMILES string: standardization and neutralization of charges, adjustment to pH 7 protonation rules, addition of explicit hydrogens, and 3D conformer generation with RDKit's ETKDG method (Riniker & Landrum, 2015). Receptor proteins were hydrogenated with REDUCE (v4.13) using the `FLIP` option (Asparagine (Asn), Glutamine (Gln), Histidine (His) side-chain flips) (Word et al., 1999; Chen et al., 2010). For AutoDock-family tools, inputs were converted to PDBQT with MEEKO (v0.4.0) (Forli Lab, CCSB et al.).

## C    EXTENDED RESULTS

### C.1    BLIND DOCKING

#### C.1.1    DOCKGEN RESULTS

We report blind docking success rates for ASTEX, POSEBUSTERS V2 and PDBBIND test sets in the main text in Section 4.1. In Figure 8, we present the results for DOCKGEN test set, which

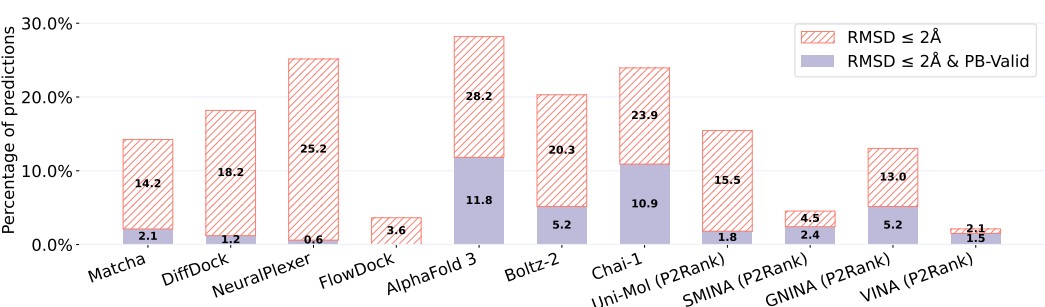

**Figure 8:** Blind ligand docking success rates on DOCKGEN dataset ($n = 330$).

contains proteins with pockets that are structurally dissimilar to the training set. All considered models perform poorly on this set. However, co-folding models show better results. We explain it by the difference in training datasets: co-folding models have seen significantly more proteins during pre-training, which allows them to work better on the out-of-distribution proteins and ligands from the DOCKGEN dataset.

### C.1.2 RESULTS WITH DIFFERENT POCKET PREDICTION METHODS

In this section, we report results for blind docking scenario for models that require pocket information as input: UNI-MOL, SMINA, VINA, GNINA. We use three types of pocket information: P2RANK (Krivák & Hoksza, 2018), FPOCKET (Le Guilloux et al., 2009) and full protein setup. Full protein means providing the whole protein and the protein center as a starting pocket center: using large box centered on the protein with padding `autobox_add 16`. The full protein setup was not used for UNI-MOL because this model is unable to process such type of inputs.

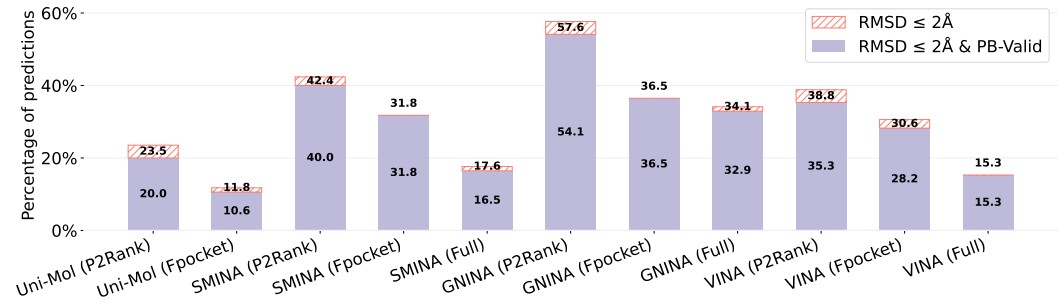

**Figure 9:** Blind ligand docking success rates with different pocket prediction methods on ASTEX Diverse set.

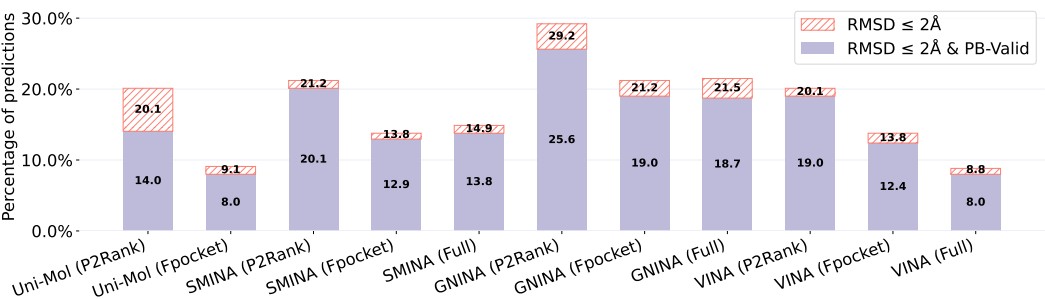

**Figure 10:** Blind ligand docking success rates with different pocket prediction methods on PDBBIND test set.

We report the results for all four considered tests datasets in Figures 9, 10, 11, and 12. According to them, P2RANK consistently shows the best quality among all other pocket identification strategies.

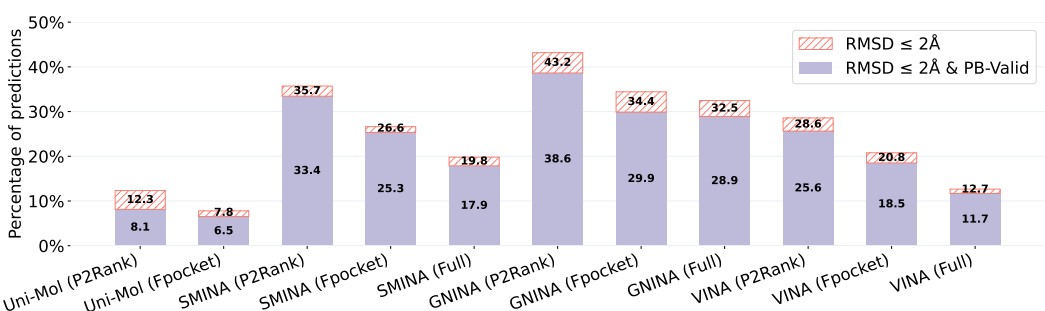

**Figure 11:** Blind ligand docking success rates with different pocket prediction methods on POSEBUSTERS V2 dataset.

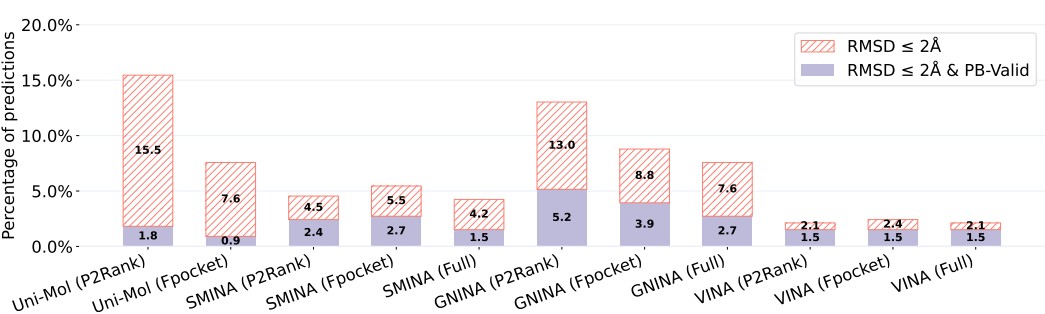

**Figure 12:** Blind ligand docking success rates with different pocket prediction methods on DOCKGEN dataset.

## C.2 POCKET-AWARE DOCKING

In addition to the blind docking setup, we run all models in the pocket-aware scenario, providing them with information about the true binding site (true ligand center). This scenario imitates the real-world case with the desired pocket for the protein provided. However, UNI-MOL cannot be fairly compared in the pocket-aware scenario due to its pocket cutting approach. Unlike other docking methods that can flexibly utilize pocket information as a starting point, UNI-MOL uses the provided reference pocket center to cut a small fixed-radius pocket around the ligand. This tight spatial constraint creates information leakage about the true ligand binding location, as the model becomes unable to "forget" or deviate significantly from the provided center due to the severely limited protein context. In contrast, methods like SMINA, VINA, GNINA, and MATCHA treat pocket information as a flexible starting point: they can explore and modify the binding site during their search process. Therefore, we do not report UNI-MOL results in the pocket-aware scenario.

In pocket-aware scenario with a known pocket center (stage 2 and stage 3 of MATCHA), we outperform classical docking tools on ASTEX, PDBBIND, DOCKGEN, but have slightly lower scores on POSEBUSTERS V2 due to the domain shift (Figures 13, 14, 15, and 16).

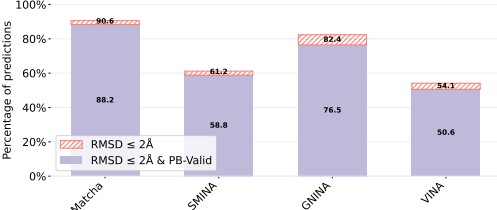

**Figure 13:** Pocket-aware ligand docking success rates on ASTEX Diverse set.

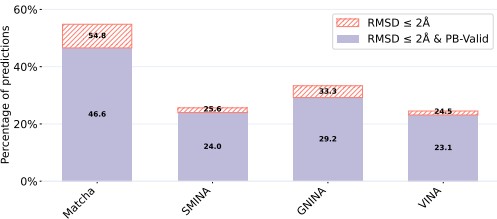

**Figure 14:** Pocket-aware ligand docking success rates on PDBBIND test set.

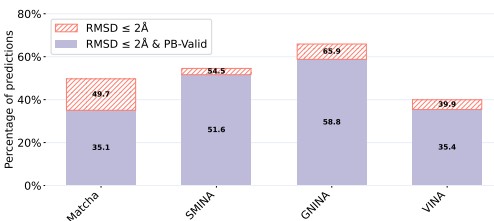

**Figure 15:** Pocket-aware ligand docking success rates on POSEBUSTERS V2 set.

**Figure 16:** Pocket-aware ligand docking success rates on DOCKGEN test set.

# D ABLATION STUDIES

## D.1 PIPELINE ABLATION

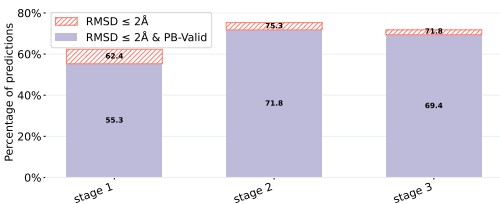

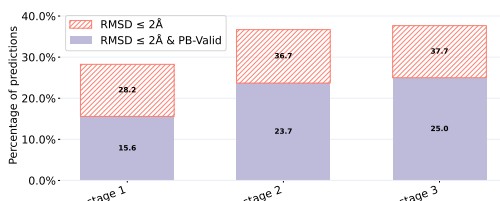

**Figure 17:** Blind ligand docking success rates after each pipeline step on ASTEX Diverse set.

**Figure 18:** Blind ligand docking success rates after each pipeline step on POSEBUSTERS V2 dataset.

We report the results produced by each pipeline stage to demonstrate the effectiveness of a three-stage pipeline design and the improvements produced by each pipeline stage. The results are reported for ASTEX and POSEBUSTERS V2 datasets in Figures 17 and 18. The results demonstrate the effectiveness of a three-stage pipeline design: most complexes are successfully docked within the first two stages. Stage 3 is an additional low-noise refinement step that provides small improvements on complex datasets such as POSEBUSTERS V2.

## D.2 NUMBER OF INFERENCE STEPS

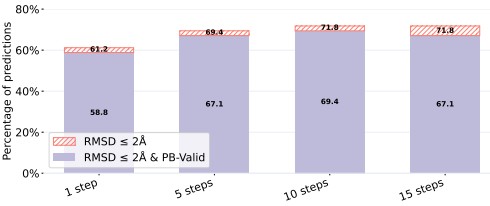

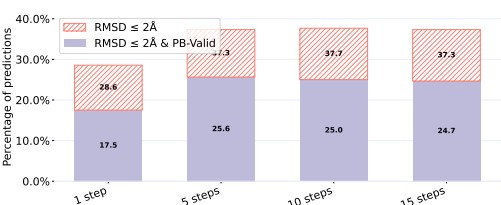

**Figure 19:** The dependence between the blind ligand docking success rates and number of Euler steps on ASTEX Diverse set.

**Figure 20:** The dependence between the blind ligand docking success rates and number of Euler steps on POSEBUSTERS V2 dataset.

We conducted experiments to define the optimal number of Euler steps during inference. The results are reported for POSEBUSTERS V2 dataset in Figures 19 and 20. Increasing the number of Euler steps from 1 to 5 and then to 10 yields clear improvements in RMSD $\leq 2$ Å and RMSD $\leq 2$ Å & PB-valid. Beyond 10 steps, the gains saturate, and 15 steps do not provide measurable improvement.

# E COMPARISON TO NEW BASELINES

We computed docking quality metrics for two new baseline methods: DYNAMICBIND (Lu et al., 2024) and SURFDOCK (Cao et al., 2025), which use the equivariant geometric diffusion networks.

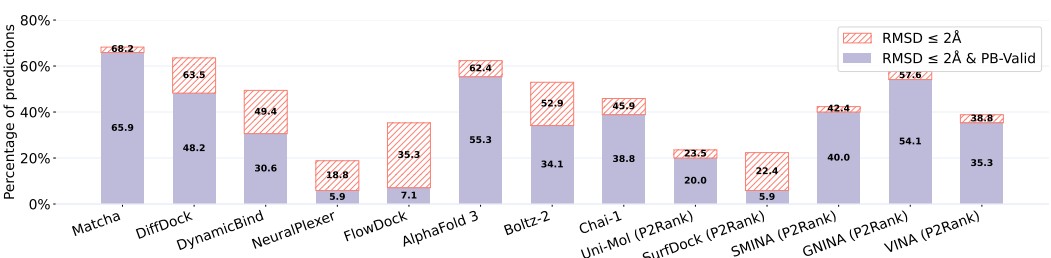

**Figure 21:** Blind ligand docking success rates on ASTEX Diverse set.

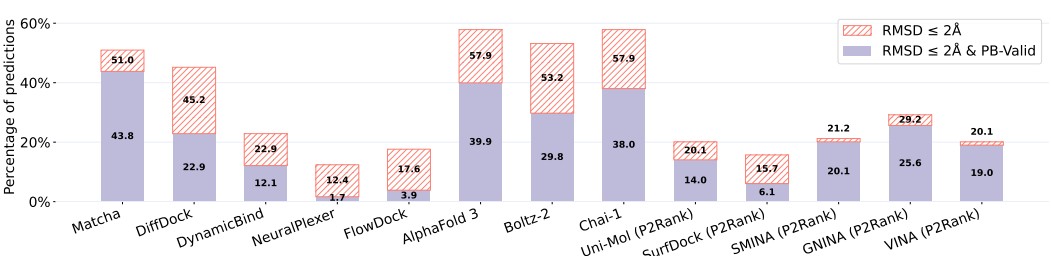

**Figure 22:** Blind ligand docking success rates on PDBBIND test set.

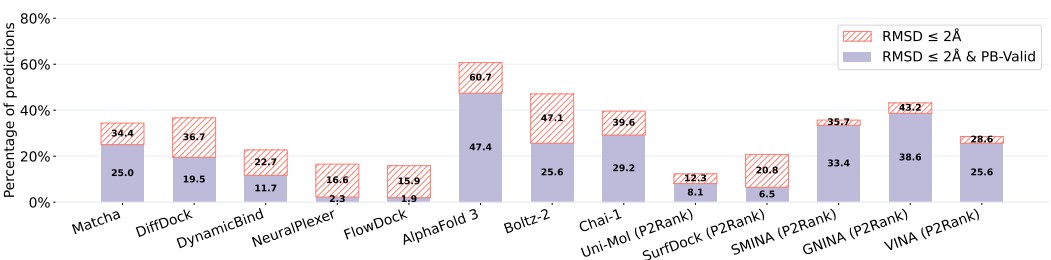

**Figure 23:** Blind ligand docking success rates on POSEBUSTERS V2 dataset.

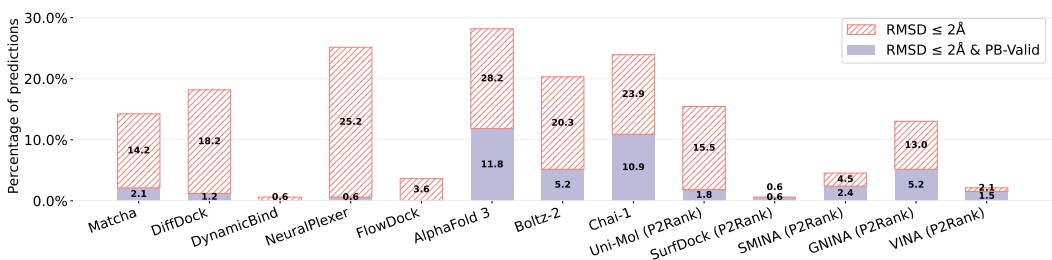

**Figure 24:** Blind ligand docking success rates on DOCKGEN dataset.

Since SURFDOCK is a pocket-based tool, we follow the same evaluation protocol as we used previously and use P2RANK to identify pockets. For both SURFDOCK and DYNAMICBIND we use experimentally-determined protein structures to be consistent with all previous evaluations. The results for all considered test datasets are shown in Figures 21, 22, 23, and 24. Across all test sets and both metrics ($\mathrm{RMSD} \leq 2\,\text{Å}$ and $\mathrm{RMSD} \leq 2\,\text{Å}$ & PB-valid), MATCHA consistently outperforms both DYNAMICBIND and SURFDOCK. We also compare inference speed (see Figure 6): the average runtime per complex is approximately 15 s for MATCHA, 91 s for DYNAMICBIND, and 73 s for SURFDOCK. Thus, MATCHA is not only more accurate on these benchmarks, but also considerably faster at inference.

## F   BENCHMARKING ON UNSEEN POSEBUSTERS SPLIT

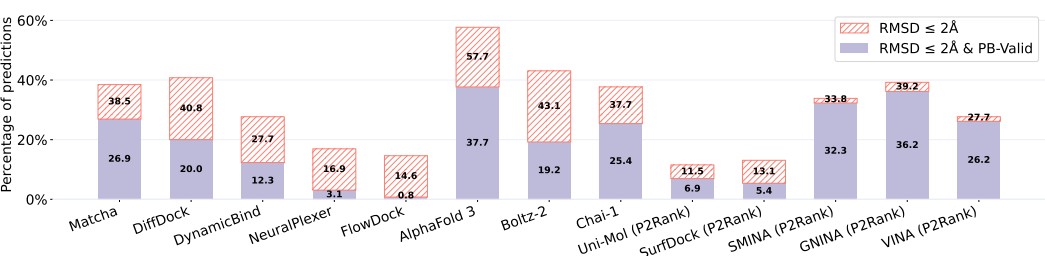

**Figure 25:** Blind ligand docking success rates on the subset of POSEBUSTERS V2 dataset with 130 / 308 complexes.

Following POSEBENCH (Morehead et al., 2025), we exclude POSEBUSTERS V2 complexes that were present in the ALPHAFOLD3 training set. It is caused by the fact that ALPHAFOLD3 and other co-folding models were trained on September 30, 2021 cutoff of PDBBIND, while POSEBUSTERS V2 uses complexes after 2019. Our model was trained on 2019 cutoff, and we used all 308 complexes for testing. Figure 25 presents results on the subset of POSEBUSTERS V2 (n=130) which was deposited in the PDB after September 30, 2021. According to them, co-folding models show decrease in performance (up to 10% in $\mathrm{RMSD} \leq 2\,\text{Å}$ & PB-valid) compared to the quality on the full (n=308) set (see Figure 23). At the same time, MATCHA and other methods, both DL-based and classical docking, keep the performance at the same level. This fact indicates overoptimistic results for co-folding models shown on POSEBUSTERS V2 dataset due to the overlap with the training data.

## G   THE IMPORTANCE OF THE POCKET ALIGNMENT

### G.1   POCKET-ALIGNED RMSD COMPUTATION

For all models that predict the structure of the whole complex, we follow Abramson et al. (2024) and use pocket-aligned symmetric RMSD. However, since this procedure is not clearly defined in the paper, we explain in detail how we perform the BASE pocket alignment.

1. The primary protein chain with the most atoms within 10 Å of the ligand is kept.
2. The pocket is defined as all $\mathrm{C}_\alpha$ atoms within 10 Å of any heavy atom of the reference ligand, restricted to protein backbone atoms.
3. The reference pocket is aligned to the *whole predicted protein structure* by $\mathrm{C}_\alpha$ atoms in PyMOL (DeLano et al., 2002) with *five refinement cycles*, which is the default parameter.

An alternative approach to compute pocket-aware RMSD was described in Qiao et al. (2024a). This POCKET-BASED approach shares the first two stages, but then the procedure differs:

3. The predicted pocket is defined as all $\mathrm{C}_\alpha$ atoms within 10 Å of any heavy atom of the predicted ligand, restricted to protein backbone atoms.
4. *Each chain in the predicted pocket* is aligned to the reference pocket, and the chain with the minimum alignment RMSD is selected. The alignment is performed with *zero refinement cycles*.

We believe the base approach provides fair evaluation, while the pocket-based approach produces overoptimistic results due to a fundamental flaw: for multi-chain proteins with multiple binding sites it can artificially align non-corresponding pockets with low RMSD by chance. This allows predicted ligands to appear correctly positioned even when docked to entirely wrong pockets.

The pocket-based alignment artificially constrains translation error since pockets are pre-aligned, masking true docking failures that would be evident in blind docking scenarios. In contrast, real-world docking can produce large translation errors when ligands bind to incorrect sites—a critical failure mode that pocket-based metrics cannot detect.

This methodological difference explains the discrepancy between our metrics and those reported for ALPHAFOLD 3 in the NEURALPLEXER 3 (Qiao et al., 2024a) paper on the POSEBUSTERS V2 dataset. To demonstrate this bias, we show that rigid docking approaches can achieve the same artificial metric improvements when evaluated using pocket-based alignment. We demonstrate the comparison between the base and the pocket-based approaches in Appendix G.2.

## G.2 DOCKING RESULTS FOR DIFFERENT WAYS OF COMPUTING POCKET-ALIGNED RMSD

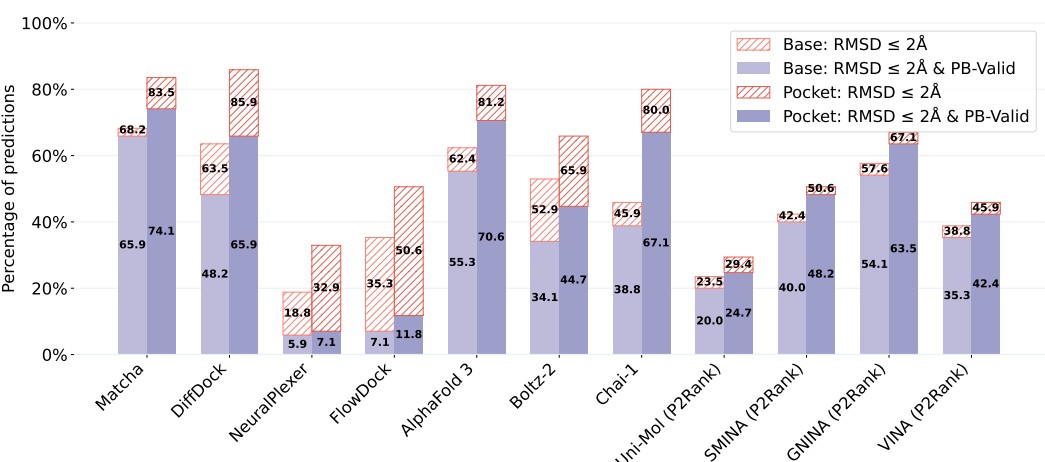

**Figure 26:** Comparison of pocket alignment strategies in blind docking scenario for ASTEX Diverse set.

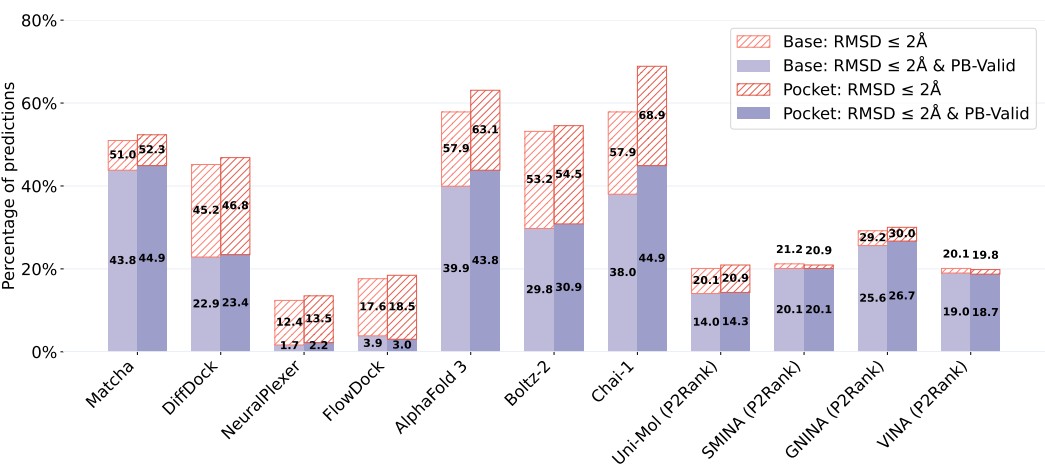

**Figure 27:** Comparison of pocket alignment strategies in blind docking scenario for PDBBIND test set.

We computed docking quality metrics using both base and pocket-based approaches for structure prediction methods (ALPHAFOLD 3, BOLTZ-2, CHAI-1, NEURALPLEXER, FLOWDOCK) and rigid docking approaches (DIFFDOCK-L, MATCHA). The results for all considered test datasets are shown in Figures 26, 27, 28, and 29. The obtained results reveal comparable metric inflation across all methods. This applies to all four test sets. The increase is around 15-20% in RMSD $\leq 2\,\text{Å}$ for the ASTEX dataset and around 10% for POSEBUSTERS V2. Lower increases for PDBBIND and DOCKGEN test sets are due to the use of the preprocessed dataset versions with the removed irrelevant chains provided by Corso et al. (2024). Moreover, the choice of the alignment almost does not affect the ordering of the docking methods: all our claims done for the base alignment, still hold for the pocket-based alignment. This demonstrates that the apparent superiority of co-folding methods in some evaluations may stem from evaluation methodology rather than genuine performance differences.

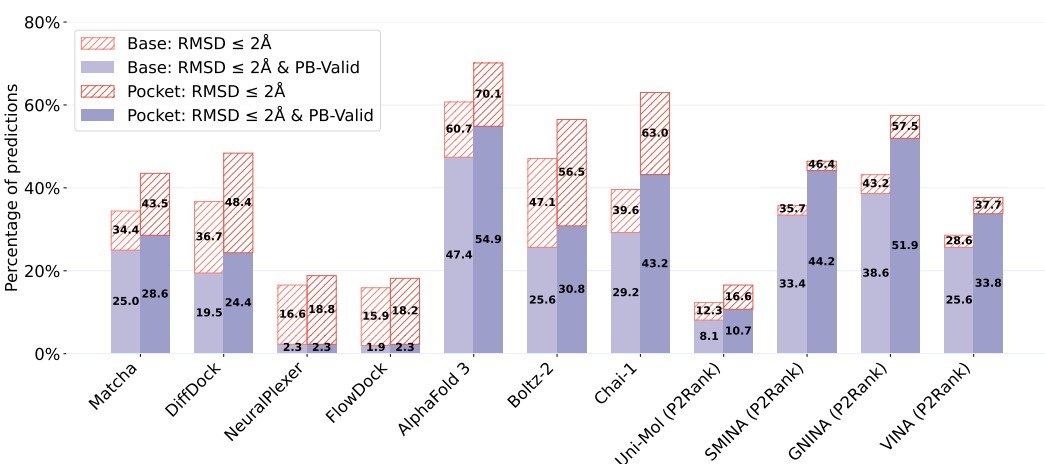

**Figure 28:** Comparison of pocket alignment strategies in blind docking scenario for POSEBUSTERS V2 dataset.

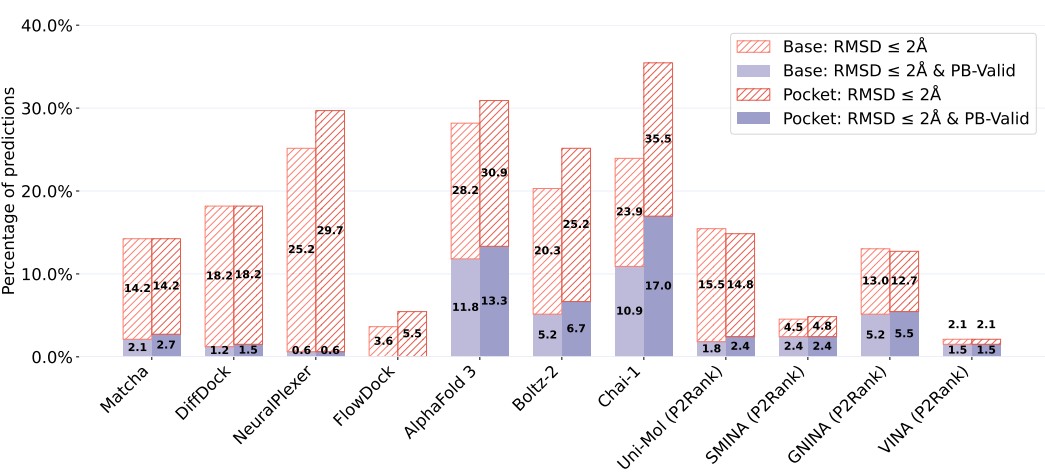

**Figure 29:** Comparison of pocket alignment strategies in blind docking scenario for DOCKGEN dataset.

## H  INFERENCE SPEED COMPARISON

**Table 1:** Comparison of the average inference time for blind docking models (for ASTEX test set)

| Method | Inference time (sec) |
|---|---|
| MATCHA | 15 |
| DIFFDOCK-L | 32 |
| NEURALPLEXER | 65 |
| FLOWDOCK | 39 |
| ALPHAFOLD 3 | 392 |
| CHAI-1 | 1638 |
| BOLTZ-2 | 1488 |

We report the average inference speed for all considered blind docking methods on one NVIDIA A100 40GB GPU. We use ASTEX dataset to measure the time on it. Time is reported only for model inference avoiding model loading. Most docking models generate multiple poses and select a pose with the best score, so we measure the time required to sample all required poses. The results are shown in Table 1.

# I PoseBusters tests

We report PoseBusters results for the following 27 tests according to the release in `https://github.com/maabuu/posebusters/releases/tag/v0.4.5`:

1. `mol_pred_loaded`,
2. `mol_cond_loaded`,
3. `sanitization`,
4. `inchi_convertible`,
5. `all_atoms_connected`,
6. `bond_lengths`,
7. `bond_angles`,
8. `internal_steric_clash`,
9. `aromatic_ring_flatness`,
10. `non-aromatic_ring_non-flatness`,
11. `double_bond_flatness`,
12. `internal_energy`,
13. `protein-ligand_maximum_distance`,
14. `minimum_distance_to_protein`,
15. `minimum_distance_to_organic_cofactors`,
16. `minimum_distance_to_inorganic_cofactors`,
17. `minimum_distance_to_waters`,
18. `volume_overlap_with_protein`,
19. `volume_overlap_with_organic_cofactors`,
20. `volume_overlap_with_inorganic_cofactors`,
21. `volume_overlap_with_waters`,
22. `double_bond_stereochemistry`,
23. `mol_true_loaded`,
24. `molecular_bonds`,
25. `molecular_formula`,
26. RMSD $\leq 2\,\text{Å}$,
27. `tetrahedral_chirality`.

