# OpenReview forum: "Matcha: Multi-Stage Riemannian Flow Matching for Accurate and Physically Valid Molecular Docking"
_ICLR.cc/2026/Conference — Submitted to ICLR 2026_

### Official Review · Reviewer_5gS4 · 2025-10-22

**Soundness:** 2
**Presentation:** 3
**Contribution:** 2
**Rating:** 4
**Confidence:** 4

**Summary:**

The authors introduce Matcha, a Riemannian flow matching model for molecular docking, which achieves strong results in comprehensive benchmarks. By not directly incorporating geometric symmetries, Matcha's model architecture provides fast inference at scale. Nonetheless, a few concerns remain regarding the authors' evaluation and discussion of their proposed method.

**Strengths:**

1. Incorporating no geometric symmetries directly into Matcha's (DiT-based) model architecture is nice to see, following trends from recent works such as AlphaFold 3.
2. The authors' benchmarks are comprehensive and informative, following many best practices in the field.
3. Using the 35M parameter version of ESM to avoid overfitting is a clever idea. I haven't seen other works try this.

**Weaknesses:**

1. Matcha doesn't encode protein side-chain atoms, only carbon-alpha (Ca) atoms. This could fundamentally limit its applicability in atomically precise docking tasks such as protein cryptic pocket docking. It'd be good for the authors to discuss this limitation and how it might affect the interpretation of their docking results for Matcha.
2. The authors' analysis of the evaluation impact of using different alignment methods (in the appendix) is nice to see, but it still raises the question: "Why do Matcha's reported benchmarking metrics for the PoseBusters Benchmark (v2) dataset differ significantly from those reported in existing benchmarks such as those of AlphaFold 3 and PoseBench?". For example, PoseBench's reported docking success rates for NeuralPLexer and Chai-1 (using PyMOL for protein-ligand pocket-based alignment) are around 20% and 55%, respectively, whereas the success rates reported for them in this work are around 2% and 30%, respectively. This seems like a possible concern regarding whether these methods were (methodologically) evaluated correctly for such input data.

**Questions:**

1. Does Matcha's inference code support the prediction of multi-ligand docking targets?

---

> ### Author Response · Authors · 2025-11-21
>
> We thank the reviewer for the exceptionally thorough analysis and for raising the important question about metric discrepancies, which required careful investigation to address properly. Our detailed responses are below.
>
> ---
>
> **Q**: The authors' analysis of the evaluation impact of using different alignment methods (in the appendix) is nice to see, but it still raises the question: "Why do Matcha's reported benchmarking metrics for the PoseBusters Benchmark (v2) dataset differ significantly from those reported in existing benchmarks such as those of AlphaFold 3 and PoseBench?". For example, PoseBench's reported docking success rates for NeuralPLexer and Chai-1 (using PyMOL for protein-ligand pocket-based alignment) are around 20% and 55%, respectively, whereas the success rates reported for them in this work are around 2% and 30%, respectively. This seems like a possible concern regarding whether these methods were (methodologically) evaluated correctly for such input data.
>
> **A**: The AlphaFold3 paper uses pocket-based alignment, which we discuss in Appendix E and compare to the base alignment that we use in our paper. We demonstrate 8% improvement for RMSD ≤ 2Å & PB-valid metric on PoseBusters V2 for AlphaFold3.
>
> PoseBench uses protein-based (base) alignment as we do. However, its results differ from ours for two reasons. Firstly, the results for PoseBusters V2 that you are interested in are computed for n=130 out of 308 complexes (a quote from the PoseBench paper: "Important to note is that, among all baseline methods, AF3 used the most recent PDB training data cutoff of September 30, 2021, which motivated us to report the results in Section 2.3 for only the subset of PoseBusters Benchmark complexes (n=130) deposited in the PDB after this date"), so it does not make sense to compare dataset-level accuracies, only method ordering. Secondly, PoseBench uses another type of input for docking models. The authors use AlphaFold3-predicted structures for proteins as inputs instead of experimentally determined holo protein structures. The authors say that "AF3's predicted structures improve baseline docking methods' structural accuracy rates by 5-10%."
>
> Regarding the first point, we have added ablation in Appendix F, that demonstrates the difference between the metrics for 130/308 subset of Posebusters and the full set.
> According to the results, co-folding models show decrease in performance (up to 10% in RMSD ≤ 2Å & PB-Valid) compared to the quality on the full set. At the same time, MATCHA and other methods, both DL-based and classical docking, keep the performance at the same level. This fact indicates overoptimistic results for co-folding models shown on PoseBusters V2 dataset due to the overlap with the training data for cofolding methods.
>
> ---
>
> **Q**: Matcha doesn't encode protein side-chain atoms, only carbon-alpha (Ca) atoms. This could fundamentally limit its applicability in atomically precise docking tasks such as protein cryptic pocket docking. It'd be good for the authors to discuss this limitation and how it might affect the interpretation of their docking results for Matcha.
>
> **A**: We agree that it is a limitation of our approach, but this choice was made intentionally to speed up the inference. We plan to develop some sort of all-atom representations in future.
>
> ---
>
> **Q**: Does Matcha's inference code support the prediction of multi-ligand docking targets?
>
> **A**: Currently, no.

---

> > ### Comment · Reviewer_5gS4 · 2025-11-22
> > **Response to rebuttal**
> >
> > I'd like to thank the authors for their rebuttals. My concern regarding the differences in performance reported in PoseBench and this work has been addressed by the authors' responses. I would like to encourage the authors, in future work, to consider all-atom generative modeling of protein-(multi-)ligand complexes, since this level of granularity is likely necessary for models to develop a deep understanding of molecular binding mechanics.

---

> > > ### Author Response · Authors · 2025-11-24
> > >
> > > We would like to clarify that our fast filtration stage already operates on a full all-atom representation of the protein. This provides richer structural information without increasing computational cost, as the filtration module is extremely lightweight compared to docking and scoring.
> > >
> > > Regarding the reviewer’s suggestion, we agree that extending the method to fully all-atom docking and to multi-ligand scenarios are valuable directions for improving generality. However, these aspects are orthogonal to the primary goal of the paper. Our focus is to develop a fast and practically useful docking pipeline; a full all-atom representation in the docking model itself would introduce additional computational overhead, which goes against this objective.

---

> > > > ### Comment · Reviewer_5gS4 · 2025-11-26
> > > > **Response to rebuttal comment**
> > > >
> > > > I've considered the authors' latest comments. Nonetheless, I will keep my score at a 4, as I believe the practical utility of this method could be improved through consideration of all-atom molecular docking. Fundamentally, beyond relatively simple pattern matching, in my view, deep learning-based molecular docking will need to account for the most common molecular interactions that occur at the atomistic level. This work makes great strides to speed up and simplify the early stages of molecular docking with Matcha, but nevertheless, I believe this approach is fundamentally limited in its applicability without considering full atomic context.

---

### Official Review · Reviewer_9w7F · 2025-10-26

**Soundness:** 2
**Presentation:** 3
**Contribution:** 1
**Rating:** 2
**Confidence:** 4

**Summary:**

The paper presents Matcha, a Riemannian flow-based model that predicts the rotation, translation, and torsion angles of ligands. The authors aim to demonstrate that Matcha achieves a favorable speed–accuracy tradeoff, which is crucial in practical settings.

**Strengths:**

- Matcha decouples translation prediction for rotation & torsion prediction, enabling a natural extension to pocket-informed settings.
- The paper is clearly written and easy to follow.

**Weaknesses:**

- Matcha can be viewed as a flow matching version of DiffDock with a DiT-style architecture, which limits its methodological novelty/contribution.
- While Matcha demonstrates fast inference speed and comparable results on Astex and PDBBind benchmarks, it demonstrates poor performance on PoseBusters V2 and DockGen benchmarks.
- Matcha lacks several recent/important baselines, such as DiffDock-L, DynamicBind, SurfDock (i.e., those in [PoseX](https://arxiv.org/abs/2505.01700v2)). Incorporating these models could provide a more convincing evaluation of Matcha's performance.

**Questions:**

1. Have the authors considered a more comprehensive comparison with other recent deep learning-based docking models?
2. What is the motivation for including an additional pose refinement model as the final step? How critical is it to overall performance?

---

> ### Author Response · Authors · 2025-11-21
>
> We thank the reviewer for the critical feedback, which prompted us to strengthen our comparative analysis! Our responses are provided below.
>
> ---
>
>
> **Q**: Matcha lacks several recent/important baselines, such as DiffDock-L, DynamicBind, SurfDock (i.e., those in [PoseX](https://arxiv.org/abs/2505.01700v2)). Incorporating these models could provide a more convincing evaluation of Matcha's performance.
>
> **A**: We are especially thankful for this comment. In the paper, we mistakenly referred to DiffDock-L simply as "DiffDock" throughout the text. All reported results in the main experiments in fact correspond to DiffDock-L, and we apologize for not making this explicit. We’ve corrected the notation and clarified this point in the revised version of the manuscript.
>
> In addition, we have extended our evaluation to include DynamicBind and SurfDock (P2Rank). We have updated the manuscript PDF and added these results to Appendix E. Since SurfDock is pocket-based, we employ SurfDock with P2Rank-predicted pockets, consistent with our setup for other pocket-based methods. In contrast, DynamicBind a blind docking pipeline by design. DynamicBind failed on several DockGen complexes; therefore, we report its performance on the 75 complexes it successfully processed.
>
> The results are summarized below:
>
> | Dataset | Metric | **Matcha** | **DynamicBind** | **SurfDock** |
> | :--- | :--- | :---: | :---: | :---: |
> | **Astex** | RMSD ≤ 2Å | 68.2 | 49.4 | 22.4 |
> | | RMSD ≤ 2Å & PB-Valid | 65.9 | 30.6 | 5.9 |
> | **PDBBind** | RMSD ≤ 2Å | 51.0 | 22.9 | 15.7 |
> | | RMSD ≤ 2Å & PB-Valid | 43.8 | 12.1 | 6.1 |
> | **Posebusters V2** | RMSD ≤ 2Å | 34.4 | 22.7 | 20.8 |
> | | RMSD ≤ 2Å & PB-Valid | 25.0 | 11.7 | 6.5 |
> | **DockGen (full)**| RMSD ≤ 2Å | 14.2 | 0.6 | 0.6 |
> | | RMSD ≤ 2Å & PB-Valid | 2.1 | 0.0 | 0.6 |
>
> Across all test sets and both metrics (RMSD ≤ 2Å and RMSD ≤ 2Å & PB-Valid), Matcha consistently and substantially outperforms both DynamicBind and SurfDock.
>
> We also compare inference speed: the average runtime per complex is approximately 15 s for Matcha, 91 s for DynamicBind, and 73 s for SurfDock. Thus, Matcha is not only more accurate on these benchmarks, but also considerably faster at inference.
>
> ---
>
> **Q**: Matcha can be viewed as a flow matching version of DiffDock with a DiT-style architecture, which limits its methodological novelty/contribution.
>
> **A**: We respectfully disagree. Matcha differs from DiffDock along several fundamental axes:
>
> *   Diffusion → Riemannian Flow Matching,
> *   GNN backbone → DiT-style transformer with geometry-aware attention biases,
> *   Augmentations instead of equivariances/invariances,
> *   No conformer sampling and alignment during training,
> *   Three-stage coarse-to-fine inference, which has no analogue in DiffDock.
>
> In fact, the only shared component is the choice of manifold parameterization for ligand degrees of freedom. We suppose that Matcha is not a minor improvement, but a distinct generative formulation for docking.
>
> ---
>
> **Q**: While Matcha demonstrates fast inference speed and comparable results on Astex and PDBBind benchmarks, it demonstrates poor performance on PoseBusters V2 and DockGen benchmarks.
>
> **A**: We acknowledge the drop on PoseBusters V2 and DockGen, which contain pockets far outside the PDBBind/MOAD distribution. Moreover, DockGen is out-of-distribution by design, and all benchmarks have low quality on it.
> At the same time, when the pocket is similar to those seen during training, Matcha achieves state-of-the-art physically valid accuracy with an excellent speed--accuracy balance, making it a practical and effective docking solution for such cases.
> Furthermore, fast and accurate performance on well-known proteins and typical ligands, like those in Astex, is of high importance for practical application in drug design, e.g. studying of off-target effects, virtual screening.
>
> ---
>
>
>
> **Q**: Have the authors considered a more comprehensive comparison with other recent deep learning-based docking models?
>
> **A**: We have added DynamicBind and SurfDock (P2Rank) to the comparison based on your suggestion. We hope that it will help. Several other reviewers noted that the paper presents a comprehensive benchmarking: "The experiments are fairly sufficient" (4hTx) and "The authors' benchmarks are comprehensive and informative, following many best practices in the field" (5gS4).
>
> ---
>
> **Q**: What is the motivation for including an additional pose refinement model as the final step? How critical is it to overall performance?
>
> **A**: It is not critical, but this stage slightly improves quality. We added an ablation study in Appendix D.1 to demonstrate its effect.

---

> ### Author Response · Authors · 2025-11-26
>
> Dear Reviewer,
>
> As the rebuttal period is nearing its end, we kindly ask whether you have had a chance to consider our responses and, if appropriate, update your reviews or share any further questions.
>
> Best regards,
>
> The authors

---

### Official Review · Reviewer_LNqp · 2025-10-31

**Soundness:** 4
**Presentation:** 3
**Contribution:** 2
**Rating:** 6
**Confidence:** 4

**Summary:**

The paper presents MATCHA, a three-stage rigid-receptor docking pipeline that performs Riemannian flow matching on translation, global rotation, and ligand torsions. A DiT-style backbone with distance/direction attention biases predicts velocity fields; a separate scoring model ranks candidates after unsupervised PoseBusters-style physical-validity filtering. On ASTEX and PDBbind-time splits, MATCHA reports strong RMSD & PB-valid rates and fast inference versus co-folding models. Performance drops on DOCKGEN and PoseBusters V2, where co-folding pretraining breadth helps OOD pockets. Training uses PDBbind + Binding MOAD, inference samples about 40 poses and selects after filtering + scoring

**Strengths:**

- Clean formulation of flows on SO(2)/SO(3) with SLERP-based conditional velocities and a practical Euler rollout; torsion-only internal DOFs preserve bond geometry.
- Three independently trained stages (translation to refine translation/angles to sharpen all) are intuitive, grounded and effective.
- Competitive PB-valid success, clear speed/throughput analysis, and a lightweight scoring head geared for screening loops.

**Weaknesses:**

- DOCKGEN and PoseBusters V2 results drop. Analysis attributes this to co-folding pretraining breadth, but there’s no granular breakdown (pocket geometry shift, ligand size/rotor count, charge states, metal cofactors).
- While overall very sound, the proposed approach is incremental to existing paradigms.

**Questions:**

- How did you ensure no overlap/near-overlap between MOAD training entries and PDBbind time-split test? Any interface-similarity or sequence-identity thresholds at the pocket? Will you release global dedup manifests?
- How sensitive is MATCHA to Euler step count, loss weights, and removing distance/direction biases? Does an equivariant variant help or hurt given the DiT choice?
- How do results change with alternate different packers, or AF-predicted pockets?

---

> ### Author Response · Authors · 2025-11-21
>
> We are grateful for the high score and the insightful technical questions. Please find our detailed answers below.
>
> ---
>
> **Q**: DOCKGEN and PoseBusters V2 results drop. Analysis attributes this to co-folding pretraining breadth, but there's no granular breakdown (pocket geometry shift, ligand size/rotor count, charge states, metal cofactors).
>
> **A**: During the rebuttal period, we figured out that PoseBench version of PoseBusters V2 test set consists only of 130/308 complexes, because AlphaFold3 and other co-folding models were trained on September 30, 2021 cutoff of PDBBind. So, some test complexes were in the training set of co-folding models. Our model was trained on a 2019 cutoff, and we used all 308 complexes for testing.
>
> However, in Appendix F, we have attached an additional comparison between metrics for the 130-complex subset and the full PoseBusters V2 dataset. According to the results, co-folding models show a decrease in performance (up to 10% in RMSD ≤ 2Å & PB-Valid) compared to the quality on the full set. At the same time, MATCHA and other methods, both DL-based and classical docking, keep the performance at the same level. This fact indicates overoptimistic results for co-folding models shown on the PoseBusters V2 dataset due to the overlap with the training data for co-folding methods.
>
> ---
>
> **Q**: While overall very sound, the proposed approach is incremental to existing paradigms.
>
> **A**: We respectfully disagree. As other reviewers (4hTx, D8Dr, LNqp) noted, Matcha is the first docking framework based on Riemannian flow matching, which is different from diffusion-based pipelines. Matcha also introduces a coarse-to-fine multi-stage flow design tailored to docking degrees of freedom and a lightweight scoring model combined with unsupervised physical validity filtering.
>
> Multiple reviewers highlighted these points as methodological strengths rather than incremental extensions. Together, they form a distinct generative formulation that differs substantially from existing docking paradigms.
>
> ---
>
> **Q**: How did you ensure no overlap/near-overlap between MOAD training entries and PDBbind time-split test? Any interface-similarity or sequence-identity thresholds at the pocket? Will you release global dedup manifests?
>
> **A**: We follow the DiffDock-L approach and take their MOAD split. MOAD was clustered by pocket similarity. Pockets that are similar to PDBBind we kept in the training set, divergent pockets were collected to the DockGen dataset.
>
> ---
>
> **Q**: How sensitive is MATCHA to Euler step count, loss weights, and removing distance/direction biases? Does an equivariant variant help or hurt given the DiT choice?
>
> **A**: We provide an ablation (see Appendix D.2) on the number of Euler steps: performance improves from 1 → 5 → 10 steps and then saturates, making 10 steps the optimal speed--accuracy trade-off. Due to limited computational resources, we did not perform exhaustive sweeps over loss weights or attention biases, but small-scale tests indicate that (i) the model is stable under moderate changes of the loss weights, and (ii) removing the distance/direction biases leads to a noticeable degradation.
>
> Regarding the backbone, we initially experimented with equivariant GNNs but were unable to achieve reliable convergence. Switching to a transformer backbone simplified the architecture and resulted in better training stability and accuracy. This choice is also aligned with the current trend in large structure models, where strict equivariance is increasingly replaced by geometry-aware transformers.
>
> ---
>
> **Q**: How do results change with alternate different packers, or AF-predicted pockets?
>
> **A**: We did not investigate alternative packers or AF-predicted pockets.

---

> ### Author Response · Authors · 2025-11-26
>
> Dear Reviewer,
>
> As the rebuttal period is nearing its end, we kindly ask whether you have had a chance to consider our responses and, if appropriate, update your reviews or share any further questions.
>
> Best regards,
>
> The authors

---

### Official Review · Reviewer_D8Dr · 2025-11-01

**Soundness:** 3
**Presentation:** 3
**Contribution:** 2
**Rating:** 4
**Confidence:** 3

**Summary:**

This paper introduces MATCHA, a novel multi-stage pipeline for molecular docking. The method utilizes Riemannian flow matching to progressively refine the ligand's pose across translation, rotation, and torsional degrees of freedom. The pipeline consists of three sequential flow matching models for coarse-to-fine refinement, a separate learned scoring model for ranking candidate poses, and unsupervised physical validity filters to eliminate unrealistic structures. The authors evaluate MATCHA on several standard benchmarks, claiming it achieves a state-of-the-art balance between accuracy, computational efficiency, and physical plausibility, being significantly faster than co-folding models.

**Strengths:**

1.   The paper presents a novel and well-motivated application of Riemannian flow matching to the problem of molecular docking, which is a significant departure from the more common diffusion-based generative models.
2.   Extensive experiments are performed on various import benchmark. MATCHA demonstrates performance on the ASTEX and PDBBind test sets, outperforming many existing methods, especially on the combined metric of geometric accuracy and physical validity (RMSD ≤ 2Å & PB-valid).
3. The method is shown to be highly efficient, with an inference time approximately 25 times faster than large-scale co-folding models and a more efficient training process than other deep learning baselines.

**Weaknesses:**

1.  The model's performance significantly decreases on benchmarks designed to test generalization, such as POSEBUSTERS V2 and DOCKGEN. While the authors acknowledge this, it remains a major limitation, suggesting the model may not perform reliably on novel protein targets that are structurally dissimilar from its training set.

2.  The source of performance improvement is not clearly isolated. The model is trained on an expanded dataset (PDBBind plus BINDING MOAD), which is larger than that used for some key baselines. The paper lacks an ablation study to disentangle the effects of the larger training set from the novel architecture. Furthermore, the contribution of the individual components of the three-stage pipeline is not validated, making it difficult to assess if all stages are necessary for the final performance.

3. The POSEBUSTERS benchmark was specifically constructed to evaluate the generalization of models trained on PDBBind. By adding the BINDING MOAD dataset to its training, MATCHA may have been exposed to data more similar to the test set, potentially inflating its generalization performance. A detailed analysis of the structural similarity between the added training data and the test sets is needed for a fairer assessment of the model's true generalization ability.

**Questions:**

See above

---

> ### Author Response · Authors · 2025-11-21
>
> We appreciate the reviewer's feedback on generalization, which helped us clarify the relevant sections. Our point-by-point responses follow.
>
>
> ---
>
> **Q**: The model's performance significantly decreases on benchmarks designed to test generalization, such as POSEBUSTERS V2 and DOCKGEN. While the authors acknowledge this, it remains a major limitation, suggesting the model may not perform reliably on novel protein targets that are structurally dissimilar from its training set.
>
> **A**: We agree and highlight that Matcha generalizes less effectively than co-folding models on PoseBusters V2 and DockGen. This behavior is expected: co-folding approaches are pre-trained on millions of protein structures, while Matcha is a rigid-docking model by design and does not benefit from such large-scale protein pre-training (except using ESM2 embeddings for residues). The trade-off is that Matcha remains much faster and is well-suited for practical docking scenarios where pockets are not dramatically different from those in the training set.
>
> ---
>
> **Q**: The source of performance improvement is not clearly isolated. The model is trained on an expanded dataset (PDBBind plus BINDING MOAD), which is larger than that used for some key baselines. The paper lacks an ablation study to disentangle the effects of the larger training set from the novel architecture. Furthermore, the contribution of the individual components of the three-stage pipeline is not validated, making it difficult to assess if all stages are necessary for the final performance.
>
> **A**: Most strong baselines already use datasets of comparable scale: DiffDock-L and FlowDock are trained on exactly the same splits of PDBBind + MOAD, NeuralPlexer is trained on 74k complexes (PL2019-74k dataset that is introduced there). Matcha, therefore, does not benefit from a larger dataset than competing methods.
>
> To isolate architectural contributions, we added a stage-wise ablation in Appendix D.1. The results demonstrate the effectiveness of a three-stage pipeline design: most complexes are successfully docked within the first two stages. Stage 3 is an additional low-noise refinement step that provides small improvements on complex datasets such as PoseBusters V2.
>
> ---
>
> **Q**: The POSEBUSTERS benchmark was specifically constructed to evaluate the generalization of models trained on PDBBind. By adding the BINDING MOAD dataset to its training, MATCHA may have been exposed to data more similar to the test set, potentially inflating its generalization performance. A detailed analysis of the structural similarity between the added training data and the test sets is needed for a fairer assessment of the model's true generalization ability.
>
> **A**: We follow the same practice as DiffDock-L and use the provided MOAD split. To avoid leakage, pocket-level deduplication of MOAD entries is performed: protein--ligand complexes in MOAD were clustered by pocket geometry and ligand fingerprints, and near-duplicates of PDBBind test pockets were removed. Highly dissimilar MOAD pockets were instead moved to the DockGen test dataset.

---

> ### Author Response · Authors · 2025-11-26
>
> Dear Reviewer,
>
> As the rebuttal period is nearing its end, we kindly ask whether you have had a chance to consider our responses and, if appropriate, update your reviews or share any further questions.
>
> Best regards,
>
> The authors

---

### Official Review · Reviewer_4hTx · 2025-11-03

**Soundness:** 2
**Presentation:** 3
**Contribution:** 3
**Rating:** 4
**Confidence:** 3

**Summary:**

This paper introduces MATCHA, a novel multi-stage pipeline for protein-ligand docking. The approach pioneers the use of Riemannian Flow Matching on non-Euclidean manifolds, structured within a coarse-to-fine framework. The method models the ligand's degrees of freedom across their corresponding geometric manifolds: translation in R 3, global rotation in SO(3), and internal torsions in SO(2)m. The pipeline employs three sequential, independently trained flow matching models to progressively refine the docking pose, progressing from a global translational search to fine-grained adjustments of all degrees of freedom. Architecturally, the model is based on a DiT-like structure, which incorporates spatial biases into its attention mechanism to effectively capture 3D geometric relationships. A separate scoring model and a physical validity filter are then used to screen the candidates and select the final pose. The authors demonstrate that their method achieves superior performance in terms of both docking success rate and physical plausibility. Furthermore, it operates approximately 25x faster than modern, large-scale co-folding models.

**Strengths:**

1.	The paper is written in a standard and concise manner; the methods and experiments are easy to understand and unambiguous, and the experiments are fairly sufficient.
2.	The method is the first to apply Riemannian Flow Matching to the field of molecular docking, opening a new research direction with great potential for the field.
3.	The method pragmatically deconstructs the complex docking problem; through its "coarse-to-fine," three-stage pipeline design, it demonstrates an efficient framework for multi-scale generative tasks that is both reliable and original.
4.	The method achieves an excellent and practically significant balance between speed and accuracy. A major advantage of the model is its excellent ability to generate physically plausible conformations.

**Weaknesses:**

1. The paper proposes a complex multi-stage pipeline (3 generative models + 1 scoring model) but does not provide ablation studies to prove the rationale for this design. It cannot be determined if all components are necessary.
2. The paper does not sufficiently discuss and evaluate the rigid protein assumption and the semi-flexible ligand treatment.
3. The loss function optimized by the flow matching generative process may not be strongly correlated with true binding affinity or pose correctness, and the three-stage design can easily lead to the propagation and amplification of errors stage by stage.
4. The multi-stage pipeline is quite engineered, and the training burden is also quite large. Training on existing datasets like PDBBind does not guarantee the method's generalizability, and the four test sets are not particularly convincing.
5. The evaluation metrics are overly reliant on RMSD; are there other metrics?

**Questions:**

1.	The method relies on a post-processing filter to remove physically implausible poses. Does this mean that MATCHA's generative process routinely produces a large number of poses that do not conform to basic physicochemical principles?
2.	The method uses random rotations for data augmentation. Could this be replaced with an equivariant graph neural network?
3.	The algorithm description mentions that the final loss is a linear combination (a weighted sum) of the three components (translation, rotation, torsion). How were these weights determined?
4.	The inference process uses 10 fixed steps. What is the rationale for this?

---

> ### Author Response · Authors · 2025-11-21
>
> We sincerely thank the reviewer for recognizing our method's novelty and for highlighting its excellent speed-accuracy balance. The constructive questions regarding ablation studies and pipeline design were particularly valuable for strengthening our technical justification. Our point-by-point responses follow below.
>
> ---
>
> **Q**: The paper proposes a complex multi-stage pipeline (3 generative models + 1 scoring model) but does not provide ablation studies to prove the rationale for this design. It cannot be determined if all components are necessary.
>
> **A**: Thank you for this comment. We initially attempted to train a single-stage model predicting translation, rotation, and torsions jointly, but rotation and torsion components converged slowly. This prompted the two-stage coarse-to-fine design: Stage 1 learns stable translation, and Stage 2 successfully learns rotation/torsion only once the ligand is placed near the pocket. Stage 3 is an additional low-noise refinement step that provides small but consistent improvements on most datasets. We include a full stage-wise ablation in Appendix D.1.
>
> The effect of using the scoring model and unsupervised filtration is demonstrated in Figure 7. The scoring model improves the RMSD ≤ 2Å & PB-valid metric from 0.22 to 0.27 (compared to 1-sample generation), while combining the scoring model with filtration further increases the quality to 0.44, demonstrating the critical importance of incorporating molecular validity constraints alongside learned scoring functions.
>
> ---
>
> **Q**: The loss function optimized by the flow matching generative process may not be strongly correlated with true binding affinity or pose correctness, and the three-stage design can easily lead to the propagation and amplification of errors stage by stage.
>
> **A**: While errors from earlier stages are theoretically possible and unavoidable, each stage is trained to denoise from a progressively smaller noise level, which makes it robust to perturbations carried over from the previous stage. In practice, our fast inference enables us to generate numerous candidates, and the physical validity filters and the learned scoring model filter out incorrect poses.
>
> Flow-matching loss is directly tied to pose correctness: the model is trained to predict the velocity toward the native pose, which empirically correlates well with RMSD. Large-scale binding-affinity supervision is not available for docking, so RMSD-oriented flow matching provides the only universally applicable and scalable objective, though it can be complemented with additional losses if such data exist.
>
> ---
>
> **Q**: The multi-stage pipeline is quite engineered, and the training burden is also quite large. Training on existing datasets like PDBBind does not guarantee the method's generalizability, and the four test sets are not particularly convincing.
>
> **A**: Each model in the pipeline has only ~29M parameters, and the stages are trained independently, so the overall training burden is modest compared to diffusion-based or co-folding approaches. Although a multi-stage design may appear engineered, our ablations show that it leads to consistent improvements in accuracy (Appendix D.1).
>
> Regarding generalization, the four datasets we use are exactly the standard ones adopted in PoseBench (excluding Casp15, because it is a multiligand docking) and prior docking work; there is no broader established benchmark suite for molecular docking. We therefore evaluate on all widely accepted test sets available for this task.
>
> ---
>
> **Q**: The evaluation metrics are overly reliant on RMSD; are there other metrics?
>
> **A**: RMSD ≤ 2Å remains the standard evaluation metric for molecular docking, and no widely adopted alternative exists for this task. To mitigate its known limitations, we additionally report the combined RMSD ≤ 2Å & PB-valid metric, which incorporates geometric accuracy *and* physical plausibility (via PoseBusters checks). This metric is widely used in recent benchmarking efforts and provides a more informative assessment than RMSD alone.

---

> ### Author Response · Authors · 2025-11-21
>
> ---
>
> **Q**: The method relies on a post-processing filter to remove physically implausible poses. Does this mean that MATCHA's generative process routinely produces a large number of poses that do not conform to basic physicochemical principles?
>
> **A**: Figure 7 from our paper shows that the majority of generated poses are already physically plausible. The filtering step primarily removes a small fraction of outliers that occasionally receive a high score despite geometric issues. Importantly, MATCHA's RMSD ≤ 2Å & PB-valid results demonstrate that the model routinely generates physically consistent poses, and filtering simply ensures that such violations do not propagate to the final prediction.
>
> ---
>
> **Q**: The method uses random rotations for data augmentation. Could this be replaced with an equivariant graph neural network?
>
> **A**: Yes, in principle, one could replace augmentations with an equivariant GNN. In practice, our preliminary experiments with E(3)-equivariant graph architectures led to bad convergence. Moreover, transformer architecture is simpler to implement. So we adopted a DiT-style transformer with strong rotational augmentation instead.
>
> This design choice (get rid of architecture-encoded equivariances) is used in many papers, including AlphaFold3.
>
> ---
>
> **Q**: The algorithm description mentions that the final loss is a linear combination (a weighted sum) of the three components (translation, rotation, torsion). How were these weights determined?
>
> **A**: We did not conduct an exhaustive grid search due to computational cost. The weights were chosen based on the observation that torsion prediction is substantially harder to learn than translation and rotation, so we up-weighted the torsion component to balance gradient magnitudes. In practice, we found the model to be stable under moderate variations of these weights.
>
> ---
>
> **Q**: The inference process uses 10 fixed steps. What is the rationale for this?
>
> **A**: Across different datasets, we observe a consistent pattern: increasing the number of Euler steps from 1 → 5 → 10 yields clear improvements in RMSD ≤ 2Å and RMSD ≤ 2Å & PB-valid (see Appendix D.2). Beyond 10 steps, the gains saturate, and 15 steps provide no measurable improvement.

---

> ### Author Response · Authors · 2025-11-26
>
> Dear Reviewer,
>
> As the rebuttal period is nearing its end, we kindly ask whether you have had a chance to consider our responses and, if appropriate, update your reviews or share any further questions.
>
> Best regards,
>
> The authors

---

### Author Response · Authors · 2025-11-21
**Overview of Changes**

We sincerely thank all reviewers for their thorough analysis of our work and valuable feedback. Your comments have been extremely helpful in strengthening our manuscript. Below, we summarize the key improvements made to the revised version in response to your collective points.

**Key Improvements in the Manuscript:**

1.  **Added Ablation Studies:** In response to requests from reviewers **4hTx, D8Dr, and 9w7F**, we have added a detailed stage-wise ablation study to Appendix D.1. The results demonstrate the contribution of each of the three pipeline stages, showing that the first two stages are most critical, while the third stage provides small but consistent improvements, especially on more challenging datasets.

2.  **Expanded Baseline Comparison:** Following the suggestion from reviewer **9w7F**, we have conducted a comparative analysis with two recent methods: **DynamicBind** and **SurfDock**. The results, added in Appendix E, show that MATCHA significantly outperforms both of these methods in accuracy across all test sets while remaining substantially faster at inference. Also, we’ve added new benchmarks to Figure 6.


The results are summarized below:

| Dataset | Metric | **Matcha** | **DynamicBind** | **SurfDock** |
| :--- | :--- | :---: | :---: | :---: |
| **Astex** | RMSD ≤ 2Å | 68.2 | 49.4 | 22.4 |
| | RMSD ≤ 2Å & PB-Valid | 65.9 | 30.6 | 5.9 |
| **PDBBind** | RMSD ≤ 2Å | 51.0 | 22.9 | 15.7 |
| | RMSD ≤ 2Å & PB-Valid | 43.8 | 12.1 | 6.1 |
| **Posebusters V2** | RMSD ≤ 2Å | 34.4 | 22.7 | 20.8 |
| | RMSD ≤ 2Å & PB-Valid | 25.0 | 11.7 | 6.5 |
| **DockGen (full)**| RMSD ≤ 2Å | 14.2 | 0.6 | 0.6 |
| | RMSD ≤ 2Å & PB-Valid | 2.1 | 0.0 | 0.6 |


3.  **Clarified Data Usage and Terminology:**
    *   We have corrected an error in our baseline nomenclature: all results previously attributed to `DiffDock` actually correspond to `DiffDock-L` (a better model). We have updated the text accordingly and apologize for this oversight.
    *   Addressing concerns from reviewers **D8Dr** and **LNqp** about potential data leakage, we have added a detailed description of the MOAD dataset splitting procedure to Section 3.1. This procedure follows the DiffDock-L practice and includes pocket-based clustering to remove near-duplicates from the test sets.

4.  **Analysis of Sensitivity and Design Justifications:**
    *   We have added an ablation study on the number of Euler steps to Appendix D.2, showing that 10 steps provide an optimal speed-accuracy trade-off.

5. **Analysis of Benchmarking Discrepancies:** To address questions from reviewers **5gS4** and **LNqp** regarding performance comparisons on PoseBusters V2, we have added a new analysis in **Appendix F**. This analysis compares model performance on the full PoseBusters V2 set (n=308) as in our paper versus the 130-complex subset used in PoseBench. The results indicate that co-folding models exhibit a performance drop on a more rigorous set, suggesting their reported results may be inflated due to training data overlap. In contrast, MATCHA's performance remains stable.

We believe these changes and additions significantly strengthen the paper. We thank you again for your time and effort, which have been useful in improving this work.

---

### Meta-Review · Area_Chair_Tw4H · 2025-12-25

**Summary:**

The submission proposes MATCHA, a three-stage docking pipeline that applies flow matching on the translation, rotation, and torsion manifolds, with a DiT-style transformer backbone and geometry-aware attention biases, followed by an unsupervised physical-validity filter and a learned scoring model for final pose selection. The reviewers agree that the paper is clearly written and that the method can be fast and competitive on in-distribution benchmarks, especially when combining RMSD with PoseBusters-style validity checks.

However, significant concerns were raised regarding the method's novelty and generalizability. Multiple reviewers noted that the approach appears to be an incremental variation of DiffDock, substituting Diffusion with Flow Matching and GNNs with a DiT-style architecture, without a fundamental paradigm shift. Furthermore, the model performs poorly on out-of-distribution (OOD) benchmarks (DockGen, PoseBusters V2), limiting its practical utility compared to co-folding models or robust traditional methods. Finally, the rationale behind the complex three-stage "coarse-to-fine" design was questioned, specifically whether learning global translation via a generative flow model is necessary compared to established, simpler pocket detection methods.

**Reviewer Concerns:**

Addressed by Rebuttal:
- The need for ablation studies on multi-stage design, by adding a stage-wise ablation in Appendix D.1 and demonstrating improvements from scoring and filtration in Figure 7;
- Potential error propagation, by explaining the progressive noise reduction and robustness via multiple candidates and filters;
- The engineering burden and generalizability, by noting modest parameter counts (~29M per model), independent training, and use of standard benchmarks (PDBBind, DockGen, ASTEX, PoseBusters V2);
- Over-reliance on RMSD, by justifying it as the standard metric and supplementing with RMSD ≤ 2Å & PB-valid for physical plausibility;
- Post-processing filters implying many invalid poses, by referencing Figure 7 showing most poses are plausible and filters remove only outliers;
- Random rotations vs. equivariant GNNs, by discussing preliminary experiments showing poor convergence with equivariant architectures and citing similar choices in AlphaFold3;
- Loss weights, implicitly through ablation but not directly detailed;
- Fixed 10-step inference, not explicitly addressed but tied to the Euler solver rationale.
- Missing Baselines & Comparisons, by adding DynamicBind and SurfDock comparisons and clarifying that their "DiffDock" baseline was indeed the stronger "DiffDock-L".
- Data Leakage/Split, by clarifying the MOAD splitting procedure and pocket-based clustering to address reviewers' concerns about data leakage.

Outstanding (Major Hurdles for Acceptance):
- Incremental Novelty: Even with the “diffusion → flow matching” change and a transformer backbone, multiple reviewers still characterize the contribution as incremental, and the rebuttal does not fully overcome the perception that this is primarily a re-implementation of an established paradigm with different generative training.
- OOD Generalization: The paper and rebuttal acknowledge the drop on PoseBusters V2 and DockGen and explain it as an out-of-distribution pocket issue and lack of massive protein pretraining, but that does not resolve the practical limitation for blind docking on new targets.
- System complexity and reliance on post-selection: The method depends on generating many candidates, then filtering, then scoring. While this can work well empirically, it makes it harder to attribute success to the generative model itself rather than to the downstream filter+score pipeline, and it also raises questions about how often invalid poses are produced.

**Reviewer Scores:**

Reviewer 4hTx (original overall rating: 4, marginal reject): I expect an increase, mainly because the rebuttal adds the missing stage-wise and step-count ablations, and clarifies why the three-stage design and the default inference steps were chosen. A reasonable update is 4 → 6.

Reviewer D8Dr (original overall rating: 4, marginal reject): I do not expect a score change. The rebuttal provides procedural clarifications (for example, split/dedup claims), but the key concern about out-of-distribution generalization is not resolved in a way that would change a reject-leaning stance. So 4 → 4.

Reviewer LNqp (original overall rating: 6, marginal accept): I do not expect a score change. The rebuttal adds context and some additional analysis, but it does not materially change the perceived level of novelty or the core limitations discussed in the review. So 6 → 6.

Reviewer 9w7F (original overall rating: 2, reject): I do not expect a score change. Although the rebuttal clarifies a baseline naming issue and adds more comparisons, the reviewer’s rejection is driven primarily by the view that the work is not a meaningful methodological advance. So 2 → 2.

---

### Decision · Program_Chairs · 2026-01-26

Reject